# Self-supervised learning with application for infant cerebellum segmentation and analysis

Yue Sun [1], Limei Wang[1], Kun Gao[1], Shihui Ying[1], Weili Lin [1], Kathryn L. Humphreys [2,3], Gang Li [1], Sijie Niu[1], Mingxia Liu [1] ✉ & Li Wang [1] ✉

Accurate tissue segmentation is critical to characterize early cerebellar development in the first two postnatal years. However, challenges in tissue segmentation arising from tightly-folded cortex, low and dynamic tissue contrast, and large inter-site data heterogeneity have limited our understanding of early cerebellar development. In this paper, we propose an accurate self-supervised learning framework for infant cerebellum segmentation. We validate its accuracy using 358 subjects from three datasets. Our results suggest the first six months exhibit the most rapid and dynamic changes, with gray matter (GM) playing a dominant role in cerebellar growth over white matter (WM). We also find both GM and WM volumes are larger in males than females, and GM and WM volumes are larger in autistic males than neurotypical males. Application of our method to a larger population will fuel more cerebellar studies, ultimately advancing our comprehension of its structure and function in neurotypical and disordered development.

The first two postnatal years are an exceptionally dynamic and critical period of brain structural and functional development, with even more dramatic growth for the cerebellum (i.e., little brain) than the cerebrum. Although the cerebellum only makes up around 10% of the brain mass, it contains approximately 80% of the brain's neurons and plays an essential role in motor control[1], sensory integration[2], attention[3], language[4], and the regulation of fear and pleasure responses[5]. Cerebellar dysfunction has been implicated in several neurodevelopmental disorders, including autism[6,7], attention-deficit/hyperactivity disorder (ADHD)[8], and schizophrenia[9].

Accurate characterization of cerebellar tissues is essential for unraveling the neural and biological underpinnings of both neurotypical and disordered development. It enables the identification of potential early biomarkers and facilitates timely interventions for improved outcomes. Previous studies have revealed age and sex differences in cerebellar volume, particularly in neurotypical individuals. For instance, ref. 10 demonstrated that total cerebellar volume is consistently larger in males compared to females across ages 5 to 24 years. Similarly, ref. 11 reported smaller cerebellar volumes in females

compared to males in subjects aged 3 months to 12.7 years. Sussman et al.[12] further corroborated these findings, showing that the female cerebellum is significantly smaller than the male cerebellum based on 261 neurotypical subjects aged 4 to 18 years (p value <0.05). In the quest to identify biomarkers for developmental disorders like autism, longitudinal studies have provided valuable insights. For example, ref. 13 conducted a study on subjects aged 6–35 years and found that total cerebellar volumes in individuals with autism may exhibit inverted-U curves, which show increased volumes in young children with autism and subsequently decreased to meet the curve of neurotypical groups at 12–13 years of age. Other cerebellar features found in autistic subjects included decreased vermis volume (7.5–18.5 years[14]), increased total cerebellar volume (10–30 years[15]), increased cerebellar white matter volume (1.9–5.2 years[16]), and reduced cerebellar gray matter volume (8–13 years[17]). Further study conducted by ref. 18 indicated that autistic boys exhibited greater cerebellar white matter compared to neurotypical boys between the ages of 2 and 3 years.

Despite these significant findings, most existing studies have either focused on subjects from early childhood to young adulthood

[1]Department of Radiology and Biomedical Research Imaging Center, University of North Carolina at Chapel Hill, Chapel Hill, NC 27599, USA. [2]Department of Psychology and Human Development, Vanderbilt University, Nashville, TN 37203, USA. [3]Department of Psychiatric and Behavioral Sciences, School of Medicine, Tulane University, New Orleans, LA 70118, USA. ✉e-mail: mxliu@med.unc.edu; li_wang@med.unc.edu

or infant subjects without differentiating between gray matter and white matter in cerebellar MRIs. Consequently, our understanding of the early cerebellar growth trajectories in both neurotypical and disordered development remains highly limited, especially during the first two postnatal years. This limitation is primarily due to the considerable challenges associated with accurately segmenting infant cerebellar tissues. These challenges can be summarized into three main factors. First, manual annotation is difficult because of low tissue contrast, especially for infants younger than three months old. This results in a limited number of training subjects with manual annotations. Figure 1a illustrates typical T1-weighted MR images of the infant cerebellum acquired at around ≤3, 6, 18, and 24 months of age, as well as the corresponding tissue intensity distributions, demonstrating the low tissue contrast in neonates and

posing a great challenge for automated or manual segmentation. Second, the collaborative use of multi-domain infant images, involving images acquired from different imaging sites with varying magnetic fields, head coils, and imaging parameters, poses a significant challenge known as the domain shift problem. As reported in a 6-month infant cerebrum segmentation challenge (iSeg-2019, http://iseg2019.web.unc.edu)[19], we found that a model trained on a specific-site dataset usually performs well on testing subjects from the same site, but poorly on subjects from other sites. This problem is also present when working with data from different time points, as shown in Fig. 1a, where infants at different time points have dynamic tissue contrasts and varying data distributions. Third, the arbor vitae is a complete and folded tree-like appearance; however, due to low tissue contrast and severe partial volume effect, there are

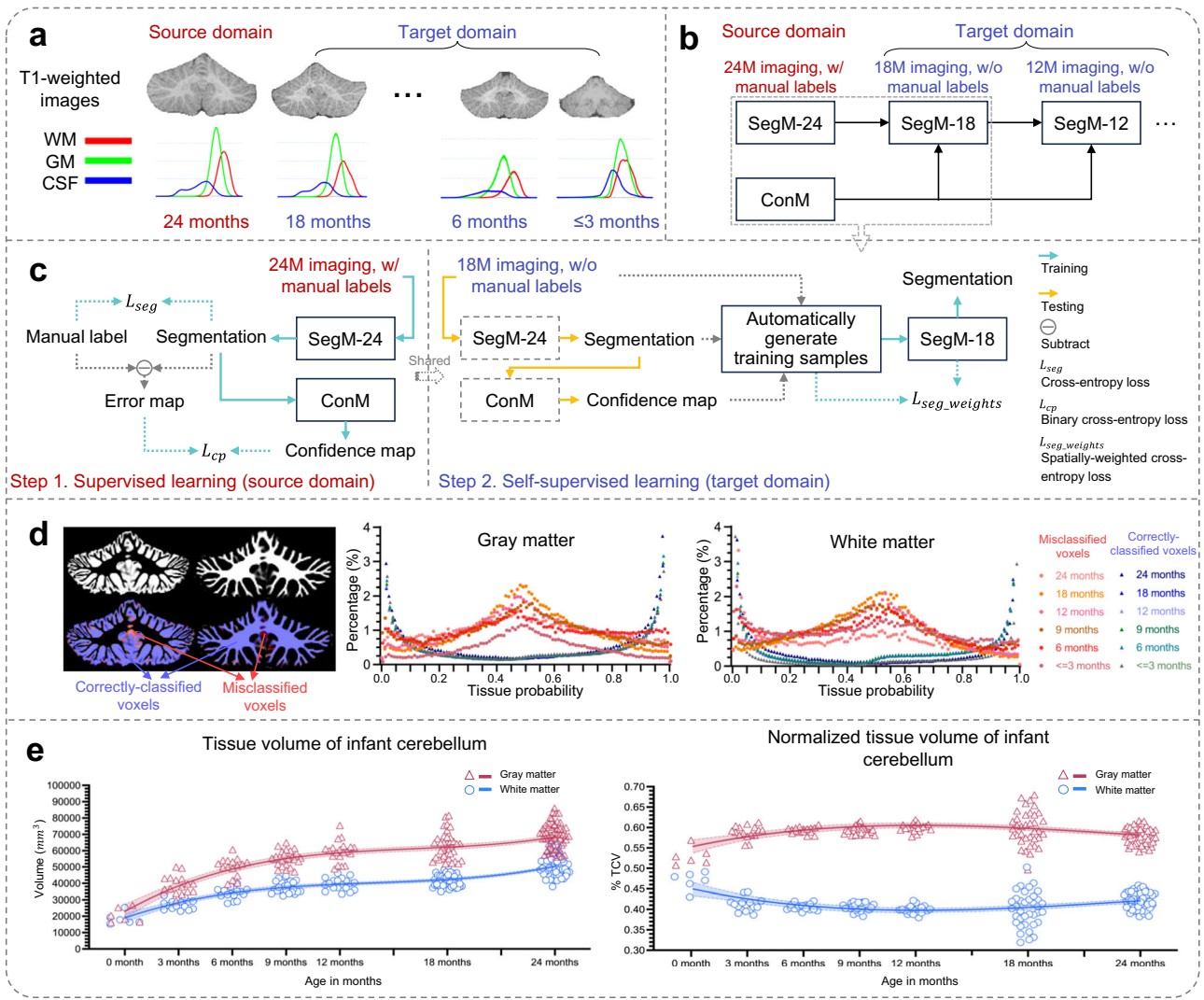

**Fig. 1 | Overview of the proposed work. a** T1-weighted MRIs of infant cerebellums and the corresponding intensity distributions in the first two postnatal years. **b** The proposed self-supervised learning (SSL) framework for infant cerebellum segmentations (24M: 24 months; 18M: 18 months). **c** The source domain comprises 24-month-old subjects with manual labels, while the target domain comprises unlabeled 18-month-old subjects. Our SSL framework consists of two steps. In Step 1, a segmentation model (ADU-Net[26]) is trained based on a number of training subjects with manual labels in the source domain. This model is then applied to testing subjects to automatically generate segmentations. Subsequently, a confidence network (with the U-Net structure[40]) is trained to evaluate the reliability of the automated segmentations at the voxel level. In Step 2, a set of reliable training

samples is automatically generated from the testing subjects to train a segmentation model for the target domain. This training is guided by a proposed spatially-weighted cross-entropy loss ($L_{seg\_weights}$). **d** Histograms of probability values for correctly classified and misclassified voxels. **e** Cross-sectional growth trajectories based on 174 neurotypical infant subjects: (left) scatterplots and fitted development trajectories (solid lines) of gray matter and white matter volumes of infant cerebellum in the first two postnatal years; (right) scatterplots and fitted development trajectories (solid lines) of normalized gray matter and white matter volumes in terms of the total cerebellum volume (TCV) in the first two postnatal years. The error bands represent the 95% confidence interval. Source data are provided as a Source Data file.

often topological errors ("hole" and "handle") in the segmentation results.

In this work, to address these challenges, we propose a self-supervised learning (SSL) framework for infant cerebellum segmentation with multi-domain data. As demonstrated in Fig. 1a, cerebellum MRIs from infants younger than 3 months old exhibit extremely low tissue contrast, posing significant challenges for manual annotations. Consequently, directly applying existing supervised learning algorithms to train a segmentation model becomes difficult. However, cerebellums MRI from 24-month-old infants exhibits much better tissue contrast[20], making it possible to segment them in a more reliable way, either manually or automatically. Therefore, we can take advantage of accurate manual labels from 24-month-old cerebellums and transfer them to other time points. To address the domain shift issue, we propose to automatically generate a set of reliable training samples for target domains and train a target domain-specific segmentation model with a spatially-weighted cross-entropy loss function. Experiments on three datasets and one challenge demonstrate the superior accuracy of our proposed framework. Note that an initial version of our framework was previously published in a conference paper[21], which, however, had limited validation on a small number of subjects from a single imaging site. In contrast, this current work presents extensive validation on multi-site subjects and introduces several key improvements. (1) We propose a novel approach of transferring manual labels from 24-month-old subjects to younger infants, enabling the generation of reliable segmentations through self-supervised learning. Moreover, we validate this method on a large number of multi-site subjects, demonstrating its effectiveness and generalizability. (2) We thoroughly evaluate the rationality of the confidence model and enhance its performance by incorporating tissue probability maps. This refinement improves the accuracy and reliability of our segmentation results. (3) We delve into early cerebellar development, examining different tissue types and sex differences. Additionally, we chart the growth trajectories of the cerebellum in both autistic and neurotypical subjects during the first two years of postnatal life. (4) We further validate our SSL framework on a multi-site infant cerebrum segmentation challenge (iSeg-2019)[19]. This additional validation demonstrates the robustness and competitiveness of our proposed framework against state-of-the-art methods.

## Results and discussion
### Experimental setup
To demonstrate the superiority of our SSL method in managing multi-site data, we verify its effectiveness on infant images from various time points (ranging from ≤3 months to 18 months) and imaging sites. In the experiments, we evaluate the performance of the proposed SSL method against several state-of-the-art methods and available pipelines for infant subjects. These include (1) volBrain[22], (2) Infant FreeSurfer[23], (3) a multi-atlas-based method[24], (4) ASD-Net[25], and (5) ADU-Net[26]. The volBrain (https://www.volbrain.net/) is an automated MRI Brain Volumetry System that uses the CERES pipeline[27] to analyze the cerebellum, which won a MICCAI cerebellum segmentation challenge. The CERES method applies a patch-based multi-atlas method, using cerebellar atlases from adults, to perform cerebellum segmentation. Infant FreeSurfer (https://surfer.nmr.mgh.harvard.edu/fswiki/infantFS) is an automated segmentation and surface extraction pipeline designed for infants. The multi-atlas-based method is a widely used label fusion segmentation framework. The ASD-Net is an attention-based semi-supervised deep learning framework that uses a generative adversarial network (GAN) to predict labels and confidence maps, and then trains the segmentation model with automatically generated labels. The ADU-Net is the backbone architecture of our segmentation model.

To ensure a fair comparison, competing methods and our SSL method are trained and tested on the same subjects. Additionally, we use the same gradual label propagation strategy as the ASD-Net method (detailed in the section "Methods"), which involves considering all testing subjects in the target domains as training samples and using a conventional cross-entropy loss to train a domain-specific segmentation model for each target domain. We evaluate the segmentation accuracy using two commonly used metrics: Dice ratio[28] and 95th percentile Hausdorff Distance (HD95)[29]. The Dice ratio measures the overlap ratio between the estimated segmentation and the manual segmentation and is a volume-based assessment. The HD95 is a surface-based assessment that measures the distance between the estimated surface and the manual surface, and the 95th percentile is used to avoid the effect of outliers. A higher Dice ratio and a lower HD95 value indicate better segmentation results.

### Cross-time point cerebellum segmentation (Siemens scanner)
We compared our SSL method with five competing methods in cross-time point cerebellum segmentation using the UNC/UMN Baby Connectome Project (BCP)[30] dataset with the Siemens Prisma scanner. The 24-month-old subjects were used as training data, while subjects at earlier time points (i.e., ≤3, 6, 9, 12, and 18 months) were used as testing data. Figure 2 presents a visual comparison of the segmentation results, where the T1w and T2w images, segmentation results generated by different methods, and corresponding manual labels are shown from top to bottom.

From the figure, it can be observed that the cerebellum segmentation achieved by our proposed SSL method are more consistent with the manual labels compared to the other methods. Specifically, the segmentation results of volBrain, Infant FreeSurfer, and the multi-atlas-based method are coarse and inaccurate, since these methods highly rely on the image registration accuracy. However, due to the low tissue contrast and convoluted folds in infant cerebellar images, it is challenging to find accurate correspondences between the atlases and individual infant cerebellum. Similarly, the ASD-Net cannot generate reasonable segmentation results. This may be because ASD-Net, with only a simple discriminator network, cannot accurately detect unreliable regions in the cerebellum with complex tissue structures. Although the ADU-Net produces better results than volBrain, Infant FreeSurfer, the multi-atlas-based method, and ASD-Net, its results for subjects less than 3 months of age are still unsatisfactory. This is because ADU-Net directly applies the model trained on 24-month-old subjects to younger subjects, ignoring the distribution gap between different time points. In terms of speed, our method is faster than other non-deep-learning methods (volBrain, Infant FreeSurfer, and the multi-atlas-based method) during the testing stage, according to Supplementary Table 4.

**Quantitative analysis.** We quantitatively compared our method with five competing methods on the BCP dataset with the Siemens Prisma scanner. Specifically, we reported the Dice ratio and HD95 of six methods on 50 infant subjects in Table 1. In addition, we performed the Wilcoxon signed-rank test (two-sided) to evaluate the statistical difference between our SSL method and each of the five competing methods. Since the volBrain and Infant FreeSurfer methods cannot well segment CSF, we did not report the corresponding results in Table 1. Table 1 shows that the proposed method consistently outperforms the five competing methods at all five time points, while the ADU-Net usually performs better than volBrain, Infant FreeSurfer, the multi-atlas-based method, and ASD-Net. Notably, the overall Dice ratio gradually decreases when segmenting younger subjects. For example, the Dice ratio for WM achieved by ADU-Net at 18 and ≤3 months of age are 92.79 and 83.96%, respectively, while those of SSL are 93.02 and 90.39%, respectively. Regarding the HD95 metric, our SSL also achieves better results than the competing methods. These results indicate that our method generates more accurate segmentation results compared with the competing methods, especially for the most

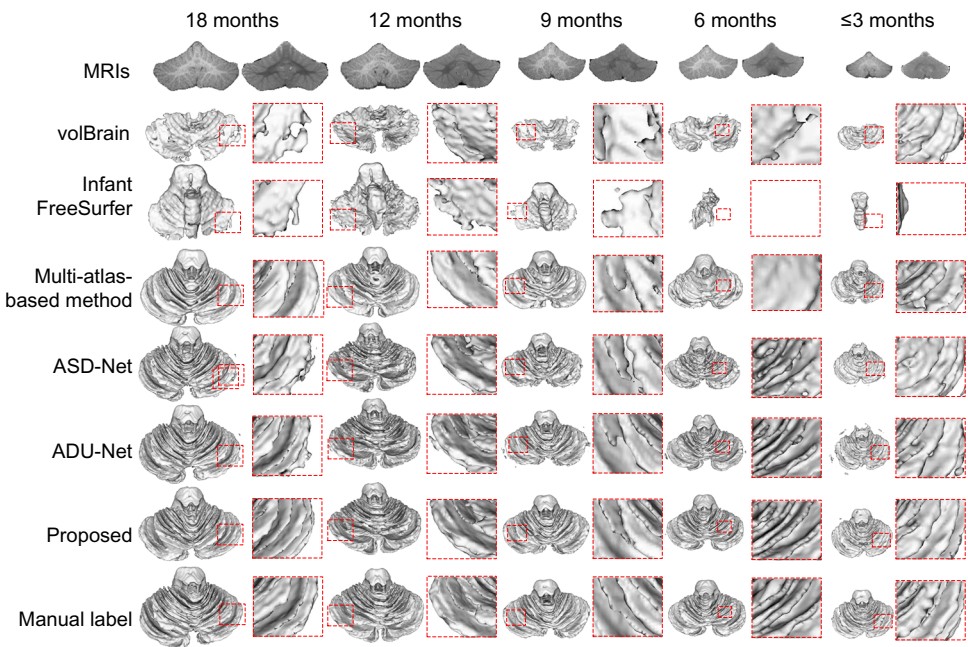

**Fig. 2 | Cerebellum segmentation comparisons for infant subjects from the BCP dataset at around 18, 12, 9, 6, and ≤3 months.** The first row displays the infant cerebellum MRIs (left: T1w images; right: T2w images) from each time point. From the second row to the seventh row, the segmentations are obtained by volBrain[22],

Infant FreeSurfer[23], a multi-atlas-based method[24], ASD-Net[25], ADU-Net[26], and our proposed SSL method. The corresponding manual labels are shown in the last row. Zoomed views are also provided for better visualization.

challenging infant subjects younger than 3 months old. More validation in the source domain (24-month-old subjects) is available in Supplementary Note 2.

**Qualitative analysis.** In addition to the quantitative evaluation on 50 subjects with manual labels, we performed a further validation of our method on 208 subjects without manual labels to demonstrate its robustness through visual inspection. Each segmentation result was visually categorized into three quality groups—"good", "fair", and "poor"—independently annotated by two raters. Figure 3 shows the number distribution of 208 subjects at different time points in a characteristic rendering of WM results of different groups in b and the assessment in each group by two raters in c. To test the inter-rater reliability, we evaluated the ratings of the two raters. The results indicated that 192 out of 208 subjects (92.31%) were rated with the same quality assessment, indicating high inter-rater reliability. From the assessment in c, our method achieved a decent average percentage of "good" and "fair" (i.e., 85.34% averaged by two raters).

**Cross-site cerebellum segmentation (Philips scanner)**
Meantime, we evaluated the performance of different methods in a cross-site cerebellum segmentation task. Specifically, we first directly applied our self-trained segmentation model for 6-month-old subjects in the subsection "Cross-time point cerebellum segmentation (Siemens scanner)" to five 6-month-old infant images acquired with a Philips scanner (the second dataset). Then, we trained a segmentation model according to our SSL strategy. For the ASD-Net, we applied its 6-month-old segmentation model in the subsection "Cross-time point cerebellum segmentation (Siemens scanner)" to the testing subjects and then trained a segmentation model with automatically generated training samples. For the ADU-Net, we directly applied the 24-month-old segmentation model in the subsection "Cross-time point cerebellum segmentation (Siemens scanner)" to the testing subjects. The visual segmentation results and quantitative evaluation in terms of the Dice ratio are shown in Fig. 4. From these results, we observed that the four competing methods (i.e., volBrain[22], Infant FreeSurfer[23], ASD-

Net[25], ADU-Net[26]) failed to generate reasonable results though the results of ASD-Net and ADU-Net are better than others. In contrast, our method generated much better results in this cross-site segmentation task, primarily due to our self-supervised learning strategy to train a site-specific segmentation model. To better compare the performance across sites, we also included the results on ten 6-month-old BCP subjects acquired with the Siemens scanner in Fig. 4b. Herein, we only quantitatively compared our results with ASD-Net and ADU-Net, since they have demonstrated much better performance than non-deep-learning methods (volBrain, Infant FreeSurfer, and the multi-atlas-based method) in Fig. 2, Table 1, and Fig. 4a. We found that the Dice ratios of our method were more consistent across the two imaging sites than the other competing methods. This demonstrates that our SSL has a good generalization ability across sites.

**Cerebellar volume analysis of neurotypical infants**
Leveraging "good" or "fair" segmentation results from 174 subjects (rated by both raters) in the section "Results and discussion", we analyzed the growth trajectory of infant cerebellums from birth to 2 years old, specifically at 0 month (2M/4F), 3 months (7M/9F), 6 months (7M/9F), 9 months (12M/14F), 12 months (6M/14F), 18 months (17M/20F), and 24 months (27M/26F), where F and M denote female and male, respectively. Figure 1e presents the scatterplots of cerebellar GM and WM volumes of infant cerebellums (left), and the scatterplots of normalized cerebellar tissue volumes in terms of the total cerebellum volume (TCV) (right). From the left figure, we observe that the GM volume increases by 237%, and the WM volume increases by 175% in the first two years of life. Furthermore, the first six months of life may be the most dynamic and critical period of cerebellum development, with volumes of (GM and WM) increasing by (141.58 and 85.18%), respectively, from 0 → 6 months, (19.92 and 15.72%) from 6 → 12 months, (6.90 and 7.06%) from 12 → 18 months, and (9.06 and 19.90%) from 18 → 24 months. The right scatterplot in Fig. 1e presents the normalized GM and WM volumes in terms of TCV to reflect the growth relative to the whole cerebellum. We observe that the percentage of GM increases significantly, particularly in the first year, whereas the WM percentage

**Table 1 | Dice ratio and 95th percentile Hausdorff distance (HD95) of cerebellum segmentation results on 50 testing subjects (N = 10 for each time point) at 18, 12, 9, 6, and ≤3 months of age from BCP (mean ± standard deviation)**

| Age (Month) | N | Method | CSF | | GM | | WM | |
|---|---|---|---|---|---|---|---|---|
| | | | Dice ratio (%) | HD95 (mm) | Dice ratio (%) | HD95 (mm) | Dice ratio (%) | HD95 (mm) |
| 18 | 10 | volBrain | N/A | N/A | 76.30 ± 3.01* | 16.37 ± 3.06 | 53.18 ± 2.67* | 25.18 ± 1.63* |
| | | Infant FreeSurfer | N/A | N/A | 71.28 ± 4.43* | 16.16 ± 2.22* | 58.24 ± 2.01* | 16.42 ± 1.95* |
| | | Multi-atlas-based method | 53.03 ± 6.28* | 11.96 ± 3.01* | 68.90 ± 6.89* | 11.41 ± 2.24 | 65.46 ± 9.06* | 8.94 ± 2.74 |
| | | ASD-Net | 73.65 ± 2.44* | 12.89 ± 1.53* | 85.19 ± 0.12* | 11.36 ± 2.77 | 88.28 ± 1.46* | 6.51 ± 3.31 |
| | | ADU-Net | 87.17 ± 6.22 | 10.46 ± 2.01 | 91.57 ± 3.10 | 13.98 ± 2.66 | 92.79 ± 2.50 | 6.56 ± 3.62 |
| | | Proposed | 89.45 ± 1.96 | 9.68 ± 1.75 | 92.12 ± 0.95 | 12.81 ± 4.45 | 93.02 ± 0.61 | 5.92 ± 2.91 |
| 12 | 10 | volBrain | N/A | N/A | 74.20 ± 8.91* | 19.92 ± 4.01 | 51.47 ± 3.37* | 22.84 ± 0.33* |
| | | Infant FreeSurfer | N/A | N/A | 67.63 ± 8.57* | 15.70 ± 3.12* | 56.39 ± 7.65* | 15.07 ± 2.02* |
| | | Multi-atlas-based method | 52.62 ± 10.70* | 14.85 ± 1.64* | 71.66 ± 3.39* | 17.10 ± 2.35* | 66.15 ± 3.71* | 8.50 ± 1.96* |
| | | ASD-Net | 81.22 ± 4.04* | 13.56 ± 1.36* | 85.05 ± 1.93* | 12.30 ± 1.62 | 84.57 ± 1.96* | 6.25 ± 0.72 |
| | | ADU-Net | 88.72 ± 9.61 | 11.67 ± 2.55 | 88.59 ± 2.68 | 14.52 ± 2.50 | 88.08 ± 1.68* | 6.81 ± 1.38 |
| | | Proposed | 90.07 ± 1.68 | 9.18 ± 2.42 | 91.66 ± 1.59 | 10.58 ± 5.14 | 92.05 ± 2.65 | 5.58 ± 1.40 |
| 9 | 10 | volBrain | N/A | N/A | 77.27 ± 4.28* | 16.89 ± 1.43 | 51.63 ± 1.21* | 21.02 ± 1.06* |
| | | Infant FreeSurfer | N/A | N/A | 69.86 ± 6.70* | 17.64 ± 4.38 | 55.15 ± 8.70* | 18.28 ± 5.96* |
| | | Multi-atlas-based method | 33.93 ± 2.44* | 17.48 ± 1.44* | 63.78 ± 0.90* | 21.88 ± 4.96* | 63.60 ± 2.30* | 9.28 ± 0.92 |
| | | ASD-Net | 76.33 ± 5.22* | 14.09 ± 1.65* | 83.30 ± 2.63* | 13.37 ± 4.43* | 84.28 ± 2.54* | 7.29 ± 2.83 |
| | | ADU-Net | 87.64 ± 6.91 | 10.81 ± 2.35 | 88.45 ± 1.65* | 16.61 ± 5.45 | 87.49 ± 2.41* | 7.80 ± 2.64 |
| | | Proposed | 87.10 ± 2.56 | 7.39 ± 2.83 | 90.96 ± 1.87 | 16.46 ± 4.29 | 92.30 ± 3.07 | 6.76 ± 3.25 |
| 6 | 10 | volBrain | N/A | N/A | 74.23 ± 6.18* | 17.78 ± 1.99 | 48.65 ± 4.87* | 19.54 ± 3.73* |
| | | Infant FreeSurfer | N/A | N/A | 67.81 ± 5.42* | 15.25 ± 1.55 | 59.78 ± 5.66* | 14.78 ± 1.41* |
| | | Multi-atlas-based method | 32.27 ± 3.19* | 17.62 ± 0.57* | 67.81 ± 5.42* | 16.52 ± 1.55* | 63.62 ± 6.15* | 9.97 ± 1.89* |
| | | ASD-Net | 71.57 ± 3.81* | 14.76 ± 2.30* | 81.14 ± 4.38* | 14.08 ± 7.91 | 83.76 ± 3.59* | 11.46 ± 9.48 |
| | | ADU-Net | 82.63 ± 7.88 | 15.69 ± 7.05 | 87.49 ± 2.52* | 16.71 ± 6.83 | 87.80 ± 1.75* | 11.53 ± 9.19 |
| | | Proposed | 84.46 ± 2.15 | 10.80 ± 1.77 | 89.52 ± 1.30 | 13.37 ± 3.05 | 91.12 ± 1.60 | 5.02 ± 0.79 |
| ≤3 | 10 | volBrain | N/A | N/A | 73.94 ± 4.95 | 18.24 ± 5.00 | 50.64 ± 2.66* | 23.22 ± 5.71* |
| | | Infant FreeSurfer | N/A | N/A | 59.40 ± 3.06* | 17.82 ± 7.22 | 26.30 ± 2.08* | 20.00 ± 1.67* |
| | | Multi-atlas-based method | 26.16 ± 9.73* | 16.37 ± 0.77* | 57.67 ± 8.26* | 16.14 ± 1.49 | 64.70 ± 13.39* | 11.45 ± 3.83* |
| | | ASD-Net | 73.64 ± 6.39* | 13.12 ± 2.50* | 77.42 ± 7.30* | 15.70 ± 9.69 | 81.71 ± 6.66* | 6.41 ± 1.17 |
| | | ADU-Net | 76.67 ± 4.81* | 10.90 ± 1.45* | 80.65 ± 8.35* | 16.96 ± 9.50 | 83.96 ± 7.61* | 7.73 ± 1.52* |
| | | Proposed | 83.87 ± 8.87 | 8.86 ± 1.02 | 87.75 ± 9.18 | 16.04 ± 10.47 | 90.39 ± 8.59 | 5.97 ± 1.06 |

The symbol "*" indicates that our proposed SSL method is significantly better than volBrain[22], Infant FreeSurfer[23], multi-atlas-based method[24], ASD-Net[25], and ADU-Net[26], with p value <0.05 (Wilcoxon signed-rank test, two-sided).

decreases. Therefore, compared with WM, GM plays a dominant role in the rapid growth of the infant cerebellum.

Figure 5a, b depicts the GM and WM volumes of the cerebellum in male and female subjects during the first two postnatal years. Note that the brain stem was not excluded when we calculated the WM/GM volume. As seen in both scatterplots (a and b), the GM and WM volumes are consistently larger in males than females from 0 to 24 months. Statistical analysis in Supplementary Table 3 indicates that cerebellar GM volumes differ significantly between male and female subjects at 6, 9, 12, 18, and 24 months (p value <0.05 based on the Wilcoxon rank-sum test or two-sample t-test, two-sided), with very large effect sizes at 6, 12, and 18 months (Cohen's $d > 1.3$), a large effect size at 9 months (Cohen's $d = 1.0541$), and medium effect sizes at the remaining months (Cohen's $d = 0.7342$ at ≤3 months; Cohen's $d = 0.6854$ at 24 months). Cerebellar WM volumes also exhibit significant differences at ≤3, 6, 12, and 18 months, as shown by both the Wilcoxon rank-sum test and two-sample t-test (p value <0.05, two-sided). These differences are accompanied by very large or large effect

sizes (Cohen's $d = 1.0883$ at ≤3 months; Cohen's $d = 2.1281$ at 6 months; Cohen's $d = 1.1847$ at 12 months; Cohen's $d = 1.4480$ at 18 months). Notably, the trajectory of infant cerebellum development may suffer from the limited number of studied subjects in this work, but it may still be worthy of reference for future work.

**Longitudinal cerebellar volume analysis of autistic infants**

With the reliable and accurate tissue segmentations generated by the proposed method, we investigated potential differences in cerebellar growth trajectories between male autistic and neurotypical subjects. Due to the limited number of female autistic subjects, we only compared autistic male subjects with neurotypical male subjects using data from the National Database for Autism Research (NDAR)[31,32] (https://nda.nih.gov/edit_collection.html?id=19/), which includes 95 longitudinal male subjects (22 autistic males and 73 neurotypical males) scanned at 6 months, 12 months, and 24 months of age. The segmentation results from all 95 × 3 = 285 scans were carefully inspected by two raters, and none of them were rated as "poor".

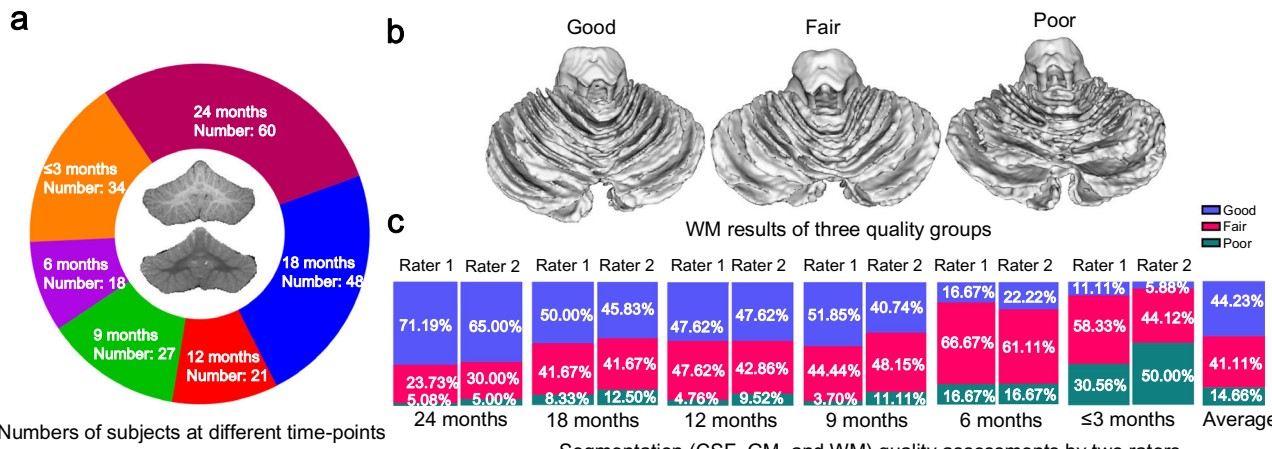

**Fig. 3 | The segmentation quality assessment on 208 subjects at different time points by two raters. a** The number distribution of 208 subjects at different time points. **b** Three representative examples for the quality of WM rendering results, i.e., "good", "fair", and "poor". **c** Assessment by two raters. Source data are provided as a Source Data file.

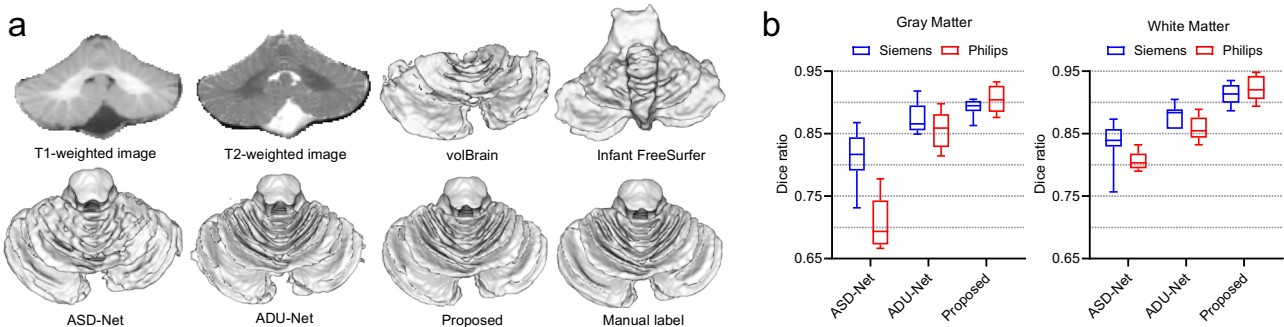

**Fig. 4 | Segmentation comparison of five methods for 6-month-old subjects acquired using Siemens and Philips scanners. a** Segmentation results on a representative 6-month-old subject acquired with a Philips scanner: T1w, T2w images, manual labels and WM segmentation results obtained by volBrain[22], Infant FreeSurfer[23], ASD-Net[25], ADU-Net[26], and the proposed method. **b** Comparison of Dice ratios of cerebellar GM and WM segmentation results between ten testing subjects from BCP (Siemens) and five testing subjects from Vanderbilt U (Philips) at 6 months of age. In each box plot, the midline represents the median value, and its lower and upper edges represent the first and third quartiles. The whiskers go down to the smallest value and up to the largest. Source data are provided as a Source Data file.

Figure 5c, d display longitudinal trajectories of GM and WM from 6 months to 24 months for both autistic and neurotypical subjects. A comprehensive list of gray and white matter volumes, along with related clinical measures for 22 autistic and 73 neurotypical subjects, is available in Supplementary Tables 5, 6. The GM volume of (autistic subjects, neurotypical subjects) increases by (23.34%, 21.17%) from 6 → 12 months, (10.29%, 11.34%) from 12 → 24 months, and (36.08%, 34.88%) from 6 → 24 months. The WM volume of (autistic subjects, neurotypical subjects) increases by (33.53%, 32.38%) from 6 → 12 months, (16.81%, 17.38%) from 12 → 24 months, and (55.97%, 55.37%) from 6 → 24 months of age. Although there is no significant difference in growth rate between the autistic and neurotypical groups (as shown in Supplementary Table 3), the GM growth rates from 6 to 12 months and from 12 to 24 months show small effect sizes between the two groups (6 → 12: Cohen's $d = 0.3473$; 12 → 24: Cohen's $d = 0.4116$). Thus, the GM and WM volumes are not only larger in autistic males compared to neurotypical males from 6 to 24 months but also have a slightly faster growth rate from 6 to 12 months. To test the statistical difference, we used the Wilcoxon rank-sum test and $t$-test to calculate the significant difference of tissue volumes between the autistic and neurotypical groups. Both tests demonstrate that the autistic GM volumes have a significant difference from those of neurotypical subjects at 12 months, with a very large effect size (Cohen's $d = 1.3151$).

We further calculated cerebellum volumes, normalized cerebellum volumes in terms of TBV, and normalized cerebellar GM and WM volumes in terms of TBV between neurotypical subjects and autistic subjects at 6 months, 12 months, and 24 months, as shown in Fig. 6. We found that as infants grow up, the difference of cerebellum volumes between neurotypical subjects and autistic infants gradually increases. Specifically, based on the Wilcoxon rank-sum test in Supplementary Table 3, the difference in cerebellum volumes is not significant at 6 months ($p$ value = 0.1647), but becomes significant at 12 months ($p$ value = 0.0162) and 24 months ($p$ value = 0.0265). However, the normalized cerebellar volumes in terms of TBV have no significant difference. In addition, for the normalized cerebellar GM and WM volumes, as shown in Fig. 6c, d, we did not find any significant difference in these trends between the neurotypical and autistic groups, but the normalized cerebellar GM volumes have small effect size at 6 and 24 months (6 months: Cohen's $d = 0.2735$; 24 months: Cohen's $d = 0.2477$), indicating that the difference between groups is not so small as to be trivial. This difference needs to be further investigated when more subjects are available in the future.

Please note that the above analysis is proof of the principle that more accurate segmentations at early ages will advance insight into the neural and biological bases. The corresponding findings should be

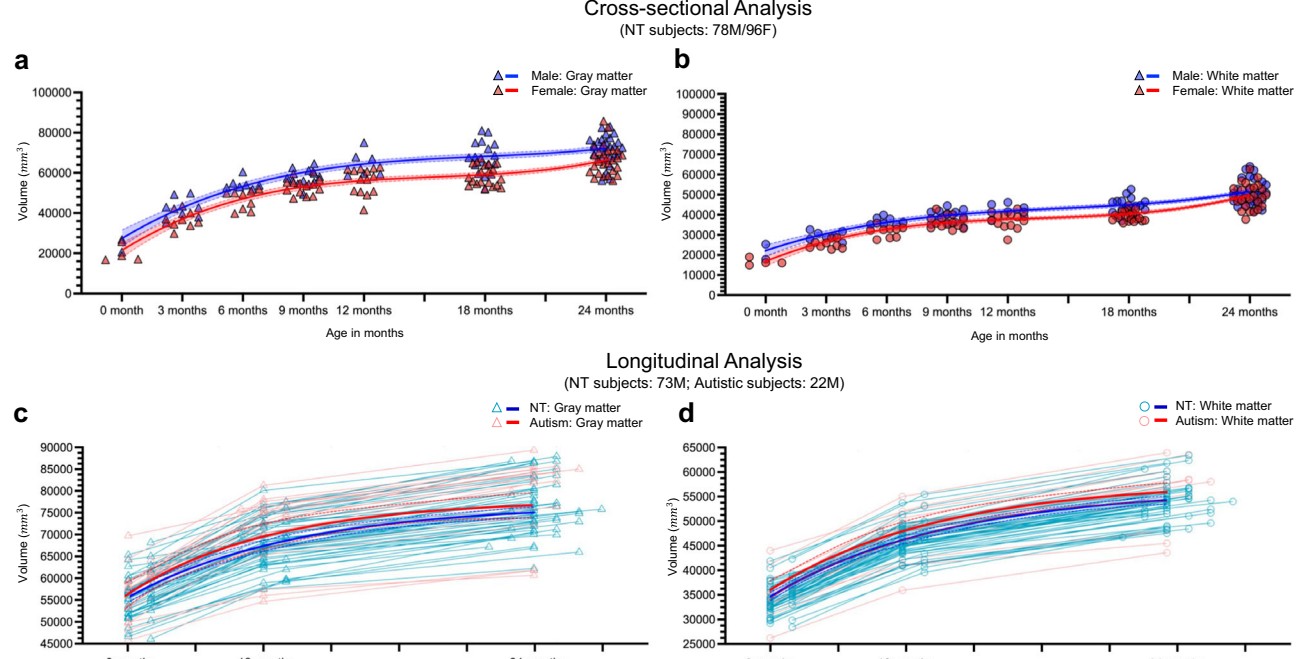

**Fig. 5 | Cross-sectional and longitudinal cerebellar analysis for neurotypical (NT) and autistic subjects in the first two postnatal years.** Cross-sectional analysis based on 174 neurotypical subjects (78M/96F): **a** Scatterplots and fitted development trajectories (solid lines) of cerebellar gray matter volumes between male and female infants in the first two postnatal years. **b** Scatterplots and fitted development trajectories (solid lines) of cerebellar white matter volumes between male and female infants in the first two postnatal years. Longitudinal analysis based on 95 male subjects from 6 months to 24 months (neurotypical subjects: 73M;

autistic subjects: 22M): **c** Longitudinal trajectories of cerebellar gray matter volumes for autistic and neurotypical males from 6 months to 24 months, and the fitted trajectories represented by solid lines. **d** Longitudinal trajectories of cerebellar white matter volumes for autistic and neurotypical males from 6 months to 24 months, and the fitted trajectories represented by solid lines. The error bands represent the 95% confidence interval. The $p$ values and effect sizes are available in Supplementary Table 3. Source data are provided as a Source Data file.

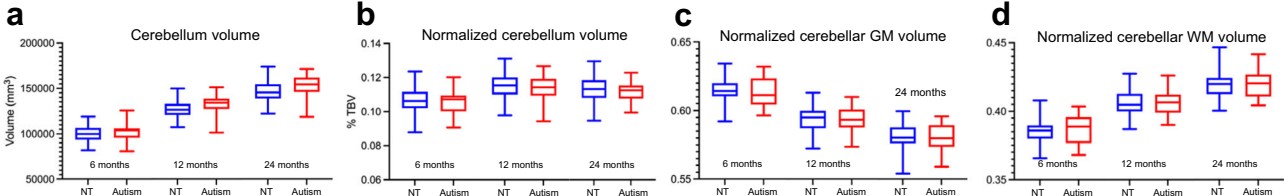

**Fig. 6 | Volume analysis for 95 male subjects from 6 months to 24 months: 73 neurotypical (NT) male subjects and 22 autistic male subjects. a** Cerebellum volumes for neurotypical and autistic infants at 6, 12, and 24 months. **b** Normalized cerebellum volumes in terms of total brain volume (TBV) for neurotypical and autistic infants at 6, 12, and 24 months. **c** Normalized cerebellar GM volumes in terms of TBV for neurotypical and autistic infants at 6, 12, and 24 months.

**d** Normalized cerebellar WM volumes in terms of TBV for neurotypical and autistic infants at 6, 12, and 24 months. In each box plot, the midline represents the median value, and its lower and upper edges represent the first and third quartiles. The whiskers go down to the smallest value and up to the largest. The $p$ values and effect sizes are available in Supplementary Table 3. Source data are provided as a Source Data file.

further validated and should not be regarded as definitive conclusions due to the limitations imposed by the available data.

### Is it easier to identify WM than GM?

The results presented in Table 1 and Fig. 4 indicate that the Dice ratio of WM obtained by the proposed method is generally higher than that of GM. This trend is also reflected in the HD95 measurements. The possible reasons for this phenomenon are twofold. *First*, GM is a narrow layer between CSF and WM, which makes it more challenging to segment as it involves identifying both the CSF/GM and GM/WM boundaries. On the other hand, segmenting WM only requires the identification of the GM/WM boundaries. Thus, identifying GM can be more challenging than identifying WM. *Second*, the tissue contrast between CSF and GM is generally lower than that between GM and WM, which can also be observed from the lower CSF accuracy. As a result, identifying the CSF/GM boundaries can be more difficult than

identifying the GM/WM boundaries, leading to lower accuracy in GM segmentation.

### Early cerebellum development

To the best of our knowledge, this might be among the first attempts to compare the GM and WM trajectories of early cerebellum development in terms of sex during the first two postnatal years. Our results indicate that the first six months of life may be the most rapid and dynamic period of cerebellum development and GM plays a dominant role in the rapid growth of cerebellum than WM. Our results also indicate that both GM and WM volumes are larger in males compared with females from 0 month to 24 months. Moreover, our longitudinal analysis in the subsection "Longitudinal cerebellar volume analysis of autistic infants" reveals abnormal growth trajectories of GM and WM for autistic boys from 6 months to 24 months of age. With limited data, our findings indicate that the GM and WM volumes are larger in autistic

boys than in neurotypical boys from 6 months to 24 months. Accurate segmentation results obtained by our method on a larger sample size may provide additional valuable information.

### Limitations and future work

Several limitations need to be considered to further improve the current framework. First, the size of the training samples remains relatively small. Collecting more neuroimaging data from multi-site MRI studies and using generative models to augment training samples is desirable. Second, the cerebellum has a fairly regular folded structure[33], than the cerebrum with highly variable folds across individuals[34]. Hence, a cerebellum atlas, providing important prior knowledge, may help guide the segmentation, especially for challenging newborn subjects. Third, the growth trajectory suffers from a limited number of cross-sectional/longitudinal subjects. As a result, the corresponding findings and conclusions need to be further validated when more subjects become available in the future. Fourth, we only segment the cerebellum into WM, GM and CSF in this work. In our future work, we will further parcellate the cerebellum into smaller regions of interest. Furthermore, modeling the appearance trajectory in terms of varying tissue contrast at different time points will also be our future work to boost the performance.

## Methods

We propose a self-supervised learning (SSL) framework for multi-site infant cerebellum MRI segmentation to characterize early cerebellar development. The framework has two steps: (1) supervised segmentation and confidence learning based on imaging data with manual labels in the source domain and (2) self-supervised segmentation learning in the to-be-analyzed target domain. In the first step, we train a segmentation model based on eighteen 24-month-old BCP subjects with manual labels, considering the high tissue contrast of MRIs at this time point. The confidence map, which is the difference between the automated segmentation predicted by the trained segmentation model and the corresponding manual label, is used to train a confidence network that predicts the reliability of the automated segmentation. In the second step, we use the confidence map predicted by the trained confidence network to generate reliable training samples for the target domain, based on which we train a domain-specific segmentation model guided by a proposed spatially-weighted cross-entropy loss. This step is performed in a self-supervised learning manner to alleviate the domain shift between different sites/time points and improve the generalization ability of the trained model. More details on the proposed SSL framework can be found in the following sections. Figure 1b, c provide an overview of the framework.

### Data and MRI preprocessing

In this study, we used T1w and T2w infant brain MRIs from three datasets, which are listed in Table 2. It should be noted that, unless otherwise defined herein, the term "image" is used interchangeably with "subject".

(1) The first dataset includes 276 subjects, with 18 labeled images used for training, and 50 labeled images and 208 unlabeled images used for validation. These images were obtained from the UNC/UMN Baby Connectome Project (BCP)[30], where images were acquired at ≤3, 6, 9, 12, 18, and 24 months of age using a Siemens Prisma scanner. Parents of all participants in BCP provide permission and informed consent prior to participation. All procedures were approved by the University of North Carolina at Chapel Hill and the University of Minnesota Institutional Review Boards. Subjects from BCP are all neurotypical subjects, and the data exclusion criteria during collection can be found in ref. 30. (2) The second dataset includes five 6-month-old MRIs that were acquired using a Philips scanner. All procedures were approved by the Vanderbilt University Institutional Review Board and all participating subjects had informed consent provided by their parent or legal guardian. These images were used to test the generalization ability of the proposed method and all competing methods. It is worth noting that a total of 55 testing subjects were used for quantitative evaluation in this study. This is currently the largest sample size used for infant cerebellum analysis, as indicated in Supplementary Table 2, which provides a summary of available cerebellum tissue segmentation methods for brain MRIs. (3) The third dataset is from the National Database for Autism Research (NDAR)[31,32] (https://nda.nih.gov/edit_collection.html?id=19/) and includes 95 male subjects. This dataset was used to investigate whether there are differences in cerebellar growth trajectories between neurotypical subjects and autistic subjects during the first two postnatal years. The infants were recruited, scanned, and accessed from four clinical data collection sites, a Data Coordinating Center, and two image processing sites (University of North Carolina at Chapel Hill, University of Washington, Children's Hospital of Philadelphia, Washington University in St. Louis, McGill University, University of Utah and UNC). The data collection sites obtained study protocol approval from their respective Institutional Review Boards, and all participating subjects had informed consent provided by their parent or legal guardian. The data exclusion criteria during collection is available in Supplementary Note 9. The diagnosis of autism was made using the DSM-IV-TR (Diagnostic and Statistical Manual of Mental Disorders, 4th Edition, Text Revision) criteria[35] at 24 or 36 months old. All images were acquired by a Siemens 3T Siemens Tim Trio scanner with a 12-channel head coil. Quality control procedures were employed to ensure image quality across different sites, times, and procedures, as described in Supplementary Note 9. More information on the subjects studied[31,32], including behavioral assessment and data exclusion criteria during collection, can be found in Supplementary Note 9. Of all the subjects available in NDAR, only 95 were longitudinally scanned at all three time points (6, 12, and 24 months of age), with 22 meeting clinical criteria for autism and 73 included as neurotypical subjects. In the Supplementary Information, Supplementary Tables 5, 6 list sex, race, and related clinical measures (e.g., Mullen[36] and ADOS[37]) for each subject. We have diligently complied with all applicable ethical regulations during the utilization of these datasets in our study.

**Table 2 | Information of three infant cerebellum datasets: (1) Baby Connectome Project (BCP) with a Siemens scanner; (2) 6-month-old MRIs acquired from Vanderbilt University with a Philips scanner; and (3) longitudinal subjects from the National Database for Autism Research (NDAR) with a Siemens scanner**

| Group | Scanner (3T) | Modality | TR/TE (*ms*) | Resolution (mm³) | Number | Age (month) |
|---|---|---|---|---|---|---|
| Training | Siemens | T1w | 2400/2.2 | 0.8 × 0.8 × 0.8 | 18 | 24 |
| | (BCP) | T2w | 3200/564 | 0.8 × 0.8 × 0.8 | | |
| Testing | Siemens | T1w | 2400/2.2 | 0.8 × 0.8 × 0.8 | 258 | ≤3, 6 |
| | (BCP) | T2w | 3200/564 | 0.8 × 0.8 × 0.8 | | 9, 12, 18 |
| | Philips | T1w | 10/4.6 | 1.0 × 1.0 × 1.0 | 5 | 6 |
| | (Vanderbilt U) | T2w | 2500/310 | 0.8 × 0.8 × 0.8 | | |
| | Siemens | T1w | 2400/3.16 | 1.0 × 1.0 × 1.0 | 95 | 6, 12, 24 |
| | (NDAR) | T2w | 3200/499 | 1.0 × 1.0 × 1.0 | | |

For image preprocessing, the resolution of all images was resampled to $0.8 \times 0.8 \times 0.8$ mm³, and T2w images were linearly aligned with their corresponding T1w images. Skull stripping and cerebellum extraction were performed using an infant cerebrum-dedicated pipeline (iBEAT V2.0[38], http://www.ibeat.cloud). Since there is no available "ground truth" segmentation for in vivo subjects, manual annotation was used as the "ground truth" for training and quantitative validation. Manual annotation was performed by a medical student (Yue Sun) trained specifically for this task and then further corrected by a medical images analysis expert (Dr. Li Wang) with 12 years of experience in infant brain MRI processing, under the guidance of a neuroradiologist (Dr. Valerie Jewells). Representative examples of manual annotations for BCP subjects are shown in Supplementary Fig. 15. The first and second columns show the original T1- and T2-weighted images, with the initial (by Yue Sun) and final annotations (by Dr. Li Wang and Dr. Valerie Jewells) shown in the third and fourth columns. The last column shows the difference between the initial annotations and the final annotations. Compared with the initial annotations, for each subject, $21{,}318 \pm 11{,}411$ voxels ($10.42 \pm 5.57\%$ of total cerebellum volume) were finally corrected by the expert and the neuroradiologist. Averagely, it took 5~6 h to perform manual annotation for a 24-month-old subject. Finally, eighteen 24-month-old subjects from BCP were manually annotated. Due to the difficulty and time-consuming nature of manual annotation editing, a limited number of younger subjects (i.e., ten subjects per time point in the first dataset and five 6-month-old subjects in the second dataset) were manually annotated for testing data, while the remaining data without manual annotations were visually inspected by two medical students (Yue Sun and Limei Wang).

### Supervised learning (source domain)

**Segmentation model at 24 months old (SegM-24).** In this work, various network architectures could be considered for supervised segmentation, including SegNet[39], U-Net[40], DenseNet[41] ADU-Net[26], and nnU-Net[42]. ADU-Net was chosen as the backbone segmentation model due to its ability to capture contextual features from global to local and its use of dense blocks to strengthen feature propagation. ADU-Net has also been successfully used in the iBEAT V2.0[38], processing over 18,000 infant cerebrum scans from 150+ institutes with various imaging protocols and scanners. ADU-Net consists of a contracting path and an expanding path, going through seven dense blocks. Each dense block includes three BN-ReLU-Conv-Dropout operations, and each convolution layer has 16 kernels with a dropout rate of 0.1. The final layer in ADU-Net is a Conv layer, followed by a softmax non-linearity to provide the per-class probability for each voxel in the MRI. As shown in Fig. 1c, a cross-entropy loss $L_{seg}$ is used in the ADU-Net, defined as

$$L_{seg} = -\sum_{i=1}^{C} Y_i \ln X_i \qquad (1)$$

where $C$ is the number of categories ($C = 4$ in this work, i.e., background, CSF, GM, and WM), $X_i$ denotes the predicted probability map, and $Y_i$ is the target of segmentation.

In this study, we fed patches from 18 BCP subjects at 24 months of age along with their corresponding manual labels into the ADU-Net. The network generated four tissue probability maps of the same size as the inputs, and the final segmentation results were determined using the softmax strategy. However, the trained segmentation model (SegM-24) could not achieve satisfactory results when directly applied to other time points or sites due to the domain shift issue. To tackle this challenge, we developed a self-supervised strategy that leverages a confidence model to automatically generate a set of reliable training samples for a target domain. This approach helps to mitigate the domain shift issue and improve the performance of the segmentation model in the target domain.

**Confidence model (ConM).** To assess the reliability of the automated segmentation at each voxel, we designed a confidence model (ConM). Since this task is relatively straightforward compared to segmentation, we use a U-Net architecture[40] for simplicity. We feed the automated segmentations and corresponding tissue probability maps as inputs to the U-Net. The confidence map, which is defined as the difference between manual labels and automated segmentations, is used as the target for training the confidence network. The confidence value $X_c \in [0, 1]$ in the confidence map is lowest (0) when the automated segmentation differs from the manual label, and highest (1) otherwise. We design a loss $L_{cp}$ to learn whether the segmentation results are reliable, defined as follows

$$L_{cp} = -(Y_c \ln X_c + a \cdot (1 - Y_c) \ln(1 - X_c)) \qquad (2)$$

where $X_c$ is the predicted confidence value, and $Y_c$ is the target. A constant parameter $a$ is empirically set as $a = 0.1$ to alleviate the volume bias between correctly classified and misclassified voxels.

In this study, we applied the ConM, which was trained on 24-month-old subjects, to all age groups. This strategy was motivated by the consistent topological errors observed in the label space across different sites and time points, as reported in previous studies[43–46]. To provide further insight into this strategy, we plotted the histogram of tissue probabilities for cerebellar WM and GM, as depicted in Fig. 1d. Our analysis revealed a distinct pattern: correctly classified voxels typically exhibited probabilities of belonging to WM or GM that were close to either 0 or 1, while misclassified voxels tended to have probabilities around 0.5. This distinction in probabilities between correctly and misclassified voxels remained consistent across different time points, motivating us to use the ConM trained on 24-month-old subjects for other age groups.

To illustrate the effectiveness of our approach, Fig. 7 shows the confidence maps and automated segmentations for 18- and 6-month-old subjects in the first and second columns, while the last column shows the corresponding manual labels. The confidence maps for both age groups were generated using the same ConM trained on 24-month-old subjects. The confidence score is represented by a color scale, where darker colors indicate lower confidence and vice versa. The 3D rendering results show that the confidence maps accurately reflect the reliability of the automated segmentation, as topological errors circled by red dashed lines in the 3D rendering are consistently reflected by dark colors in the confidence map at different ages. This is further supported by the quantitative validation performed on 45 subjects, as detailed in Supplementary Note 5.

### Self-supervised learning (target domain)

In the self-supervised learning strategy, the reliability of automated segmentation for each voxel is taken into account through the spatially-varying weights $w(x)$, which are calculated based on the confidence map,

$$w(x) = \begin{cases} M_{cp}(x), & M_{cp}(x) \geq 0.5 \\ 0, & M_{cp}(x) < 0.5 \end{cases} \qquad (3)$$

where $M_{cp}$ is the confidence map, i.e., the output of the confidence model. Voxels with higher confidence scores are assigned higher weights. We propose a spatially-weighted cross-entropy loss function that uses these weights to select reliable voxels for training in the target domain, which is defined as

$$L_{seg\_weights} = -w \sum_{i=1}^{C} Y_i \ln X_i \qquad (4)$$

With the weighted automated segmentations as pseudo-labels, we train a new segmentation model on the unlabeled subjects from the

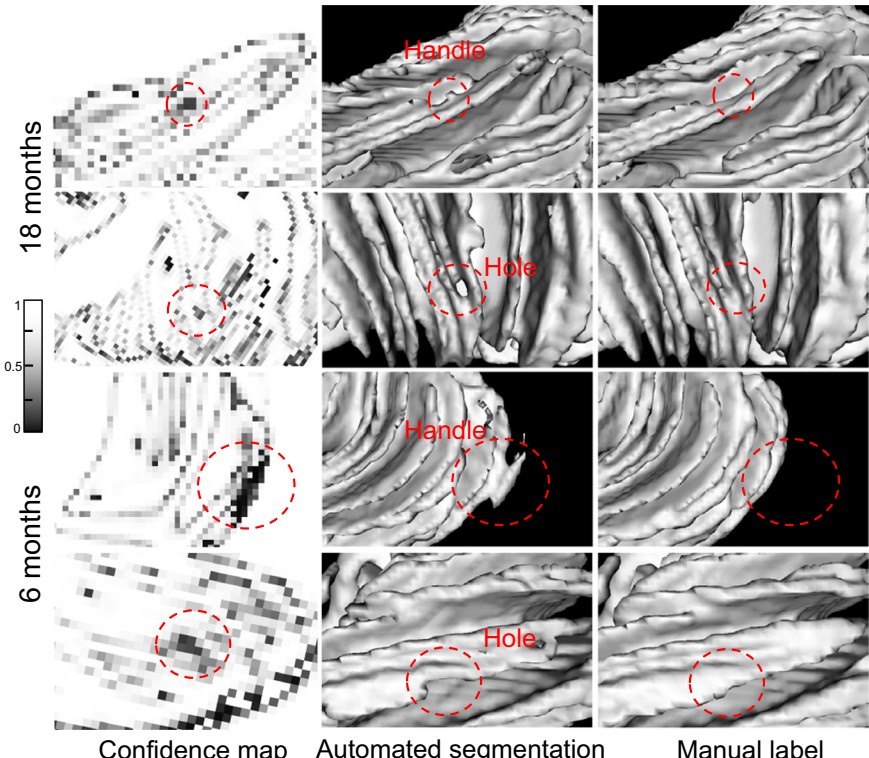

**Fig. 7 | Confidence maps and automated segmentation results at different time points.** The first column shows the confidence maps, the second column shows the segmentation results, and the last column shows the corresponding manual labels. Please note that in the confidence maps, the darker color indicates a lower confidence score and vice versa. The automated segmentations are accompanied by red dashed circles that indicate unreasonable results, such as "handle" and "hole", which are reflected in dark colors on the confidence map.

target domain. This loss function effectively supervises the training process with these reliable labels, allowing the model to learn domain-specific features and adapt to the domain shift issue. The influence of the spatially-weighted cross-entropy loss function is available in Supplementary Note 4.

It is worth noting that these reliable labels can be not only from inside the tissues, but also from the boundaries between tissues. A justification for including boundary samples in the training set can be found in Supplementary Note 8. With this approach, we can generate a set of training samples for each target domain and train a domain-specific segmentation model.

### Implementation details

We randomly extracted 1000 MR image patches (size: $32 \times 32 \times 32$) from each subject, treating the T1w and T2w images as two channels in the proposed networks. The influence of the number of training samples was explored in Supplementary Note 7. The kernels were initialized by Xavier, and we used the Stochastic Gradient Descent (SGD) strategy for network optimization. The learning rate was set to 0.005 and multiplied by 0.1 after each epoch.

To train the confidence model, we split the training subjects into $K$ folds, and trained a segmentation model based on subjects from any $K - 1$ folds, testing on subjects from the remaining fold to derive their automated segmentations and corresponding tissue probability maps. We repeated this procedure until every fold was independently tested, finally deriving automated segmentations and corresponding tissue probability maps for all subjects, which were used to train the confidence model. We set $K = 2$ for simplicity. It is important to highlight that we optimized the segmentation model and the confidence model separately. If these models were jointly trained, the segmentation model would initially exhibit a large error, making it difficult to effectively train the subsequent confidence

model. Additionally, before the convergence of the segmentation model, the target for the confidence model would constantly change, presenting significant challenges for training the confidence model. By decoupling the training process and optimizing each model independently, we were able to achieve more stable and effective results.

In the infant cerebellum segmentation task, we treated 24-month-old BCP subjects with manual labels as the source domain, and any other time points/sites as the to-be-analyzed target domain. Given the large distribution gap between 24-month-old and 0-month-old subjects, we proposed a gradual label propagation strategy ($24 \rightarrow 18 \rightarrow 12 \rightarrow 9 \rightarrow 6 \rightarrow 0 \backsim 3$). Specifically, we applied our self-supervised learning framework first to 18-month-old subjects, and regarded 18-month-old subjects with automated segmentations as a source domain for 12-month-old subjects. We repeated this procedure until reaching 0-month-old subjects. The influence of the gradual label propagation strategy is available in Supplementary Note 3.

### Reporting summary

Further information on research design is available in the Nature Portfolio Reporting Summary linked to this article.

## Data availability

The raw data from BCP[30] and NDAR[31,32] in this study are available in the NIMH Data Archive (NDA) through standard request procedures (BCP: https://nda.nih.gov/edit_collection.html?id=2848, NDAR: https://nda.nih.gov/edit_collection.html?id=19). The five Philips scans are available at https://github.com/YueSun814/Philips_data. The multi-site infant cerebrum data used in this study are available in the iSeg-2019 challenge[19] (https://iseg2019.web.unc.edu) through standard request procedures. Source data are provided with this paper.

## Code availability

The source code and training subjects with manual annotations for the proposed SSL are available online (https://github.com/DBC-Lab/Self_Supervised_Learning.git, https://zenodo.org/record/8050825). In detail, the proposed network was trained using the Caffe deep learning framework (Caffe 1.0.0-rc3). For deployment, a custom Python code (Python 2.7.17) was developed. The image preprocessing step, including skull stripping and cerebellum extraction, was performed by using a public infant-dedicated pipeline (iBEAT V2.0 Cloud[38], http://www.ibeat.cloud). CERES V1.0 pipeline[27] was used to analyze the cerebellum when submitting testing data to the volBrain website (https://www.volbrain.net). The Infant FreeSurfer pipeline (https://surfer.nmr.mgh.harvard.edu/fswiki/infantFS), updated in Feb 2020, was used to analyze testing data.

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

## Acknowledgements

Y.S., LM.Wang, and K.G. were supported by NIH grants (Nos. MH109773 to L.W. and MH117943 to L.W. and G.L.). G.L. was supported in part by NIH grants (Nos. MH117943 to L.W. and G.L., MH123202 to G.L., and MH116225 to G.L.). M.L. was supported in part by an NIH grant (No. AG073297 to M.L.). L.W. was supported in part by NIH grants (Nos. MH117943 to L.W. and G.L., MH123202 to G.L., MH116225 to G.L., AG073297 to M.L.). This work also utilizes the efforts of the UNC/UMN Baby Connectome Project Consortium. We thank Dr. Valerie Jewells for her support during the manual annotation.

## Author contributions

Y.S. and L.W. designed and implemented the pipeline. Y.S. carried out the application, performed the experiments, and analyzed the data. Y.S., LM.W., K.G. and L.W. performed result validation. W.L. and K.L.H. provided infant brain MRIs for training and testing. Y.S. wrote the manuscript. S.Y., G.L., S.N., M.L. and L.W. revised the manuscript.

## Competing interests

The authors declare no competing interests.
