## [Peer Review File · Nature Communications]

Self-Supervised Learning with Application for Infant Cerebellum Segmentation and AnalysisReviewers' comments:

Reviewer #1 (Remarks to the Author):

“The article proposes an automatic segmentation approach (see Figure 1b) for labelling the cerebellum from infant brain MRIs. The method is first trained on manual segmentations of 24 month old. Next the method learns to transfer those segmentations to younger infants. The accuracy of the method is further improved by increasing the size of the training data via automatically selecting non-manually (“unlabelled”) MRIs that the method was able to accurately segment. The approach is applied to MRIs of 333 subjects collected by three different studies showing superior accuracy over the state-of-the-art on one of the data sets. Based on automatic segmentations that were rated acceptable by human experts, they found that both cerebellum gray and white matter volumes are larger in males than females, and larger in autistic males than controlled males.

I applaud the authors for trying to tackle this extremely challenging problem. Challenges include having a large enough number of high quality (manual) segmentations for running the experiments and the relatively low contrast between tissues classes. Given their proposed approach, the knowledge of the authors regarding these issues is impressive. Furthermore, the results with respect to the autistic males are quite interesting.”

Response: We thank the reviewer for the positive comments.

Comment 1. *“However, the entire article itself lacks focus and rigor.”*

Comment 1.1. *“An example of lacking focus is Figure 1, where the connections between subfigures is entirely unclear. Choosing black and white images does not ease interpretation and the “take-home message” of that figure gets lost.”*

Response: Thanks for the valuable suggestion. In the revised manuscript, we have updated Fig. 1 by following “global-to-local” and “coarse-to-fine” rules:

- (1) Fig. 1a firstly shows the whole picture of our method by leveraging high-contrast 24-months-old subjects with annotation (source domain) to segment low-contrast 18-, 12-, ..., 0-month-old subjects without annotation (target domain). Fig. 1b further shows “how to leverage the source domain to segment the target domain”, i.e., the self-supervised learning flowchart, in which we train a segmentation model and a confidence model in the source domain, and apply them to the target domain for training a domain-dedicated segmentation model for each target domain. Finally, Fig. 1c shows the details on how to train the segmentation model and confidence model in the source domain and apply to the target domain.
- (2) We further clearly marked the training and testing steps using different colors, with legends shown in Fig. 1c.
- (3) For a better interpretation, we have updated Fig. 1e, by using light red and blue colors to denote gray and white matters (instead of the original black and white colors).

For the convenience of review, we pasted the updated Fig. 1 as follows.

Fig. 1. **a.** T1-weighted MRIs of infant cerebellums and the corresponding intensity distributions in the first two postnatal years. **b.** The proposed self-supervised learning (SSL) framework for infant cerebellum segmentations. **c.** Source domain is 24-month-old subjects with manual labels and the target domain is unlabeled 18-month-old subjects. Our SSL framework consists of two steps. In Step 1, in the source domain, a segmentation model (i.e., ADU-Net) is trained based on a number of training subjects with manual labels, and then applied to testing subjects to automatically generate segmentations. Then, a confidence network (with the U-Net structure) is trained to evaluate the reliability of those automated segmentations at the voxel level. In Step 2, a set of reliable training samples from the testing subjects is automatically generated to train a segmentation model for the target domain accordingly, guided by a proposed spatially-weighted cross-entropy loss ($L_{seg_weights}$). **d.** Histograms of probability values for correctly-classified and misclassified voxels. **e.** Cross-sectional growth trajectories based on 174 normal control infant subjects: (left) scatterplots of gray matter and white matter volumes of infant cerebellum in the first two postnatal years; (right) scatterplots of normalized gray matter and white matter volumes in terms of the total cerebellum volume (TCV) in the first two postnatal years.

Comment 1.2. “Another example is the discussion, most of which should be moved to the experiment section or omitted, such as experiments unrelated to the cerebellum (e.g. Cross-site Infant Cerebrum Segmentation).”

Response: As suggested, we have moved the Section of *Application on Cross-site Infant Cerebrum Segmentation* into the *Supplementary Materials*.

Comment 1.3. “Regarding rigor, the experimental setup is high questionable. The approach for generating manual segmentations is not specified. Was it done by one person or multiple? What was the expertise of the people involved? How was the quality of the manual generated segmentation checked? Further confusing is the description of how many subjects in each dataset were manually segmented. Table I gives some guidance but is incomplete.”

Response: Thanks for the valuable and instructive comments. In the revised manuscript, we have included the manual protocol on how to annotate the cerebellum. Manual annotation was initially performed by a medical student (Yue Sun) trained specifically for this task based on a cerebellum manual annotation protocol, and finally checked and corrected by a medical images analysis expert (Dr. Li Wang) with 12 years' experience in infant brain MRI processing, under the guidance of a neuroradiologist (Dr. Valerie Jewells). We have added new Section I of the *Supplementary Materials* to provide details of the manual protocol. We have added new Figure S13 to show representative examples of efforts of manual annotations for BCP subjects, as pasted below. The first and second columns show the original T1- and T2-weighted images, with the initial (by Yue Sun) and final annotations (by Dr. Li Wang and Dr. Valerie Jewells) shown in the third and fourth columns. The last column shows the difference between the initial annotations and the final annotations. Compared with the initial annotations, for each subject, $21,318 \pm 11,411$ voxels ($10.42\% \pm 5.57\%$ of total cerebellum volume) were finally corrected by the expert and the neuroradiologist. Averagely, it took 5~6 h to perform manual annotation for a 24-month-old subject.

Fig. S13. Efforts of manual annotation for BCP subjects from 0~3 months to 18 months. The first and second columns show the original T1- and T2-weighted images, with the initial and final annotations shown in the third and fourth columns. The last column shows the difference between the initial annotations and the final annotations. Comparing with the initial annotations, for each subject, $21,318 \pm 11,411$ voxels ($10.42\% \pm 5.57\%$ of total cerebellum volume) were finally corrected by an infant processing expert and a neuroradiologist.

For training, we have eighteen 24-months-old subjects with manual annotations. For testing, besides previous 25 testing subjects, we further have performed manual annotations on 25 new testing subjects (i.e., 5 new testing subjects for each time point), and thus we now totally have 50 ($25+25=50$) testing subjects for quantitative analysis. Accordingly, we listed the number of testing subjects with manual annotations in the second column of updated Table II, i.e., $N=10$ for each time point.

TABLE II

DICE RATIO AND 95TH-PERCENTILE HAUSDORFF DISTANCE (HD95) OF CEREBELLUM SEGMENTATION RESULTS ON 50 TESTING SUBJECTS ($N = 10$ FOR EACH MONTH) AT 18, 12, 9, 6, AND ≤ 3 MONTHS OF AGE FROM BCP (MEAN \pm STANDARD DEVIATION). THE SYMBOL “+” INDICATES THAT OUR PROPOSED SSL METHOD IS SIGNIFICANTLY BETTER THAN *volBrain* [43], *Infant FreeSurfer* [44], MULTI-ATLAS-BASED METHOD [45], ASD-NET [23], AND ADU-NET [14] (WITH p -VALUE < 0.05).

Age (Month)	N	Method	CSF		GM		WM	
			Dice ratio (%)	HD95 (mm)	Dice ratio (%)	HD95 (mm)	Dice ratio (%)	HD95 (mm)
18	10	volBrain	N/A	N/A	76.30 \pm 3.01 ⁺	16.37 \pm 3.06	53.18 \pm 2.67 ⁺	25.18 \pm 1.63 ⁺
		Infant FreeSurfer	N/A	N/A	71.28 \pm 4.43 ⁺	16.16 \pm 2.22 ⁺	58.24 \pm 2.01 ⁺	16.42 \pm 1.95 ⁺
		Multi-atlas-based method	53.03 \pm 6.28 ⁺	11.96 \pm 3.01 ⁺	68.90 \pm 6.89 ⁺	11.41 \pm 2.24	65.46 \pm 9.06 ⁺	8.94 \pm 2.74
		ASD-Net	73.65 \pm 2.44 ⁺	12.89 \pm 1.53 ⁺	85.19 \pm 0.12 ⁺	11.36 \pm 2.77	88.28 \pm 1.46 ⁺	6.51 \pm 3.31
		ADU-Net	87.17 \pm 6.22	10.46 \pm 2.01	91.57 \pm 3.10	13.98 \pm 2.66	92.79 \pm 2.50	6.56 \pm 3.62
		Proposed	89.45 \pm 1.96	9.68 \pm 1.75	92.12 \pm 0.95	12.81 \pm 4.45	93.02 \pm 0.61	5.92 \pm 2.91
12	10	volBrain	N/A	N/A	74.20 \pm 8.91 ⁺	19.92 \pm 4.01	51.47 \pm 3.37 ⁺	22.84 \pm 0.33 ⁺
		Infant FreeSurfer	N/A	N/A	67.63 \pm 8.57 ⁺	15.70 \pm 3.12 ⁺	56.39 \pm 7.65 ⁺	15.07 \pm 2.02 ⁺
		Multi-atlas-based method	52.62 \pm 10.70 ⁺	14.85 \pm 1.64 ⁺	71.66 \pm 3.39 ⁺	17.10 \pm 2.35 ⁺	66.15 \pm 3.71 ⁺	8.50 \pm 1.96 ⁺
		ASD-Net	81.22 \pm 4.04 ⁺	13.56 \pm 1.36 ⁺	85.05 \pm 1.93	12.30 \pm 1.62 ⁺	84.57 \pm 1.96 ⁺	6.25 \pm 0.72
		ADU-Net	88.72 \pm 9.61	11.67 \pm 2.55	88.59 \pm 2.68	14.52 \pm 2.50	88.08 \pm 1.68 ⁺	6.81 \pm 1.38
		Proposed	90.07 \pm 1.68	9.18 \pm 2.42	91.66 \pm 1.59	10.58 \pm 5.14	92.05 \pm 2.65	5.58 \pm 1.40
9	10	volBrain	N/A	N/A	77.27 \pm 4.28 ⁺	16.89 \pm 1.43	51.63 \pm 1.21 ⁺	21.02 \pm 1.06 ⁺
		Infant FreeSurfer	N/A	N/A	69.86 \pm 6.70 ⁺	17.64 \pm 4.38 ⁺	55.15 \pm 8.70 ⁺	18.28 \pm 5.96 ⁺
		Multi-atlas-based method	33.93 \pm 2.44 ⁺	17.48 \pm 1.44 ⁺	63.78 \pm 0.90 ⁺	21.88 \pm 4.96 ⁺	63.60 \pm 2.30 ⁺	9.28 \pm 0.92
		ASD-Net	76.33 \pm 5.22 ⁺	14.09 \pm 1.65 ⁺	83.30 \pm 2.63 ⁺	13.37 \pm 4.43 ⁺	84.28 \pm 2.54 ⁺	7.29 \pm 2.83
		ADU-Net	87.64 \pm 6.91	10.81 \pm 2.35 ⁺	88.45 \pm 1.65 ⁺	16.61 \pm 5.45	87.49 \pm 2.41 ⁺	7.80 \pm 2.64
		Proposed	87.10 \pm 2.56	7.39 \pm 2.83	90.96 \pm 1.87	16.46 \pm 4.29	92.30 \pm 3.07	6.76 \pm 3.25
6	10	volBrain	N/A	N/A	74.23 \pm 6.18 ⁺	17.78 \pm 1.99	48.65 \pm 4.87 ⁺	19.54 \pm 3.73 ⁺
		Infant FreeSurfer	N/A	N/A	67.81 \pm 5.42 ⁺	15.25 \pm 1.55	59.78 \pm 5.66 ⁺	14.78 \pm 1.41 ⁺
		Multi-atlas-based method	32.27 \pm 3.19 ⁺	17.62 \pm 0.57 ⁺	67.81 \pm 5.42 ⁺	16.52 \pm 1.55 ⁺	63.62 \pm 6.15 ⁺	9.97 \pm 1.89 ⁺
		ASD-Net	71.57 \pm 3.81 ⁺	14.76 \pm 2.30 ⁺	81.14 \pm 4.38 ⁺	14.08 \pm 7.91	83.76 \pm 3.59 ⁺	11.46 \pm 9.48
		ADU-Net	82.63 \pm 7.88	15.69 \pm 7.05 ⁺	87.49 \pm 2.52 ⁺	16.71 \pm 6.83	87.80 \pm 1.75 ⁺	11.53 \pm 9.19
		Proposed	84.46 \pm 2.15	10.80 \pm 1.77	89.52 \pm 1.30	13.37 \pm 3.05	91.12 \pm 1.60	5.02 \pm 0.79
≤ 3	10	volBrain	N/A	N/A	73.94 \pm 4.95 ⁺	18.24 \pm 5.00	50.64 \pm 2.66 ⁺	23.22 \pm 5.71 ⁺
		Infant FreeSurfer	N/A	N/A	59.40 \pm 3.06 ⁺	17.82 \pm 7.22	26.30 \pm 2.08 ⁺	20.00 \pm 1.67 ⁺
		Multi-atlas-based method	26.16 \pm 9.73 ⁺	16.37 \pm 0.77 ⁺	57.67 \pm 8.26 ⁺	16.14 \pm 1.49	64.70 \pm 13.39 ⁺	11.45 \pm 3.83 ⁺
		ASD-Net	73.64 \pm 6.39 ⁺	13.12 \pm 2.50 ⁺	77.42 \pm 7.30 ⁺	15.70 \pm 9.69	81.71 \pm 6.66 ⁺	6.41 \pm 1.17
		ADU-Net	76.67 \pm 4.81 ⁺	10.90 \pm 1.45 ⁺	80.65 \pm 08.35 ⁺	16.96 \pm 9.50	83.96 \pm 7.61 ⁺	7.73 \pm 1.52 ⁺
		Proposed	83.87 \pm 8.87	8.86 \pm 1.02	87.75 \pm 9.18	16.04 \pm 10.47	90.39 \pm 8.59	5.97 \pm 1.06

In the revised manuscript, we have added the details of the manual protocol in Section I of the *Supplementary Materials*, and the corresponding content in Section III (pasted below).

The corresponding content in Section III: “Manual annotation was initially performed by a medical student (Yue Sun) trained specifically for this task based on a cerebellum manual annotation protocol, and finally checked and corrected by a medical images analysis expert (Dr. Li Wang) with 12 years’ experience in infant brain MRI processing, under the guidance of a neuroradiologist (Dr. Valerie Jewells). Details of the manual protocol are available in Section I of *Supplementary Materials*. Figure S13 of the *Supplementary Materials* shows representative examples of efforts of manual annotations for BCP subjects. The first and second columns show the original T1- and T2-weighted images, with the initial (by Yue Sun) and final annotations (by Dr. Li Wang and Dr. Valerie Jewells) shown in the third and fourth columns. The last column shows the difference between the initial annotations and the final annotations. Compared with the initial annotations, for each subject, 21,318 \pm 11,411 voxels (10.42% \pm 5.57% of total cerebellum volume) were finally corrected by the expert and the neuroradiologist. Averagely, it took 5~6 h to perform manual annotation for a 24-month-old subject.”

Comment 1.4. “Unclear is also the modelling choices made (i.e., methodology section). For example, the authors write “ Various network architectures could be used for supervised segmentation, such as U-Net, [15], V-Net [33], U-Net++ [34], ADU-Net [14], and nnU-Net [35]. In this work, we employ ADU-Net [14] as the backbone segmentation model because it achieves descent results. ””

Response: Thanks for pointing this out. We employ ADU-Net [14] as the backbone segmentation model because it captures contextual features from global to local using a contracting path and a symmetric expanding path [15], and strengthens feature propagation by employing dense blocks [38]. For infant cerebrum segmentation [14], the ADU-Net achieves higher Dice ratios on WM, GM and CSF by 6.3%/3.1%/3.5%, 16.8%/3.3%/3.8%, and 12.7%/1.4%/1.7%, over SegNet [35]/U-Net [15]/DenseNet [36] methods, respectively. More importantly, the ADU-Net was employed in the iBEAT V2.0 (Infant Brain Extraction and Analysis Toolbox, www.ibeat.cloud), which has **successfully processed 16,000+ infant cerebrum scans from 130+ institutes with various imaging protocols and scanners**, achieving consistently better performance than the other tools. For clarity, we have revised the related text in Section IV-A of the revised manuscript:

*“In this work, we employ ADU-Net [14] as the backbone segmentation model because it captures contextual features from global to local using a contracting path and a symmetric expanding path [15], and strengthens feature propagation by employing dense blocks [38]. For infant cerebrum segmentation [14], the ADU-Net achieves higher Dice ratios on WM, GM and CSF by 6.3%/3.1%/3.5%, 16.8%/3.3%/3.8%, and 12.7%/1.4%/1.7%, over SegNet [35]/U-Net [15]/DenseNet [36] methods, respectively. More importantly, the ADU-Net was employed in the iBEAT V2.0 (Infant Brain Extraction and Analysis Toolbox, www.ibeat.cloud), which has **successfully processed 16,000+ infant cerebrum scans from 130+ institutes with various imaging protocols and scanners**, achieving consistently better performance than the other tools”.*

Comment 1.5. *“Furthermore, the word “ground truth” is confusing as it does not exist for the experimental setup.”*

Response: Thanks for the comment. We agree that there is no ground truth. For quantitative comparisons, we consider the manual annotation as the “ground truth”. To avoid the confusion, we have replaced the “ground truth” with the “manual annotation” in the revised version.

Comment 1.6. *“More importantly, modelling decisions are made based on visual evaluation of a single case (ie.. Fig 2) : “From the 3D rendering results, we can see the unreasonable results (i.e., anatomical errors) circled by red dashed lines are consistently reflected as dark colors in the confidence map at different months of age. Therefore, the same ConM (trained on 24 month-old subjects) can effectively evaluate the reliability of automated segmentations for subjects at different months of age.” It might be due to the suboptimal use of the English language but such inference is highly questionable.”*

Response: Thanks for the valuable comments. To demonstrate the advantage of the confidence model, we quantitatively compared the results with and without the confidence model on 55 testing subjects from 6 target domains in Section II of the *Supplementary Materials*. We find that the results with the confidence maps are significantly better than the results without the confidence maps (p -values < 0.05) for most age groups. Figure 2 is an illustration of the confidence map and shows how the automated segmentations are related to the reliability denoted by confidence maps. To clarify this, in the revised manuscript, we have updated the corresponding content in Section IV-A:

*“From the 3D rendering results, we can see the unreasonable results (i.e., anatomical errors) circled by red dashed lines are consistently reflected in dark colors in the confidence map at different months of age. This suggests that the confidence map defined in this work is capable to reflect the reliability of automated segmentation, as confirmed by the quantitative validation on 55 subjects in Section II of the *Supplementary Materials*.”*

Section II of the *Supplementary Materials*: *“The performance of the confidence model is critical for generating a set of reliable training samples in the target domain. Therefore, we perform a 2-fold cross-validation experiment*

(with 10 repetitions) to quantitatively analyze the performance of the confidence model, i.e., calculating the Dice ratio between predicted confidence maps and the corresponding true confidence maps for eighteen 24-month-old subjects. Specifically, in each repetition, eighteen 24-month-old subjects are randomly partitioned into two folds. Each fold is alternatively used as training subjects, while the other fold is treated as testing subjects. Fig. S8 shows the Dice ratios of the predicted confidence map in every repetition. We can see that most Dice ratio values are above 90%, which indicates the predicted confidence map can effectively detect the reliability of automated segmentations.

Fig. S8. Evaluation on predicted confidence maps on eighteen 24-month-old subjects by performing a 2-fold cross-validation with 10 repetitions.

Furthermore, to demonstrate the improvement contributed by the confidence maps, we perform an ablation experiment to compare the difference of target-domain-specific segmentation models trained with/without confidence maps. With the confidence maps, we automatically generate a set of reliable training samples, then train a target-domain-specific segmentation model for each domain with the proposed spatially-weighted cross-entropy loss. Without the confidence maps, we consider all testing data as training samples, and then leverage the cross-entropy loss to train a target-domain-specific segmentation model for each domain. There are 55 testing subjects with manual labels at 6 target domains (i.e., 50 BCP subjects at 5 time-points and 5 Philips subjects at 6 months of age); therefore, we train 6 target-domain-specific segmentation models and report the corresponding Dice ratios in Fig. S9. From Fig. S9, we can see that results with the confidence maps (red bars) are more accurate than the results without the confidence maps (blue bars). The statistical significance is listed in Table SI, from which we can see the results with the confidence maps are significantly better than the results without the confidence maps for most age groups. ”

Figure S9. Evaluation on automated segmentations for 55 cross-time-point testing subjects obtained by corresponding target-domain-specific segmentation models trained without/with confidence maps, with * indicating p-value < 0.05.

TABLE S1
P-VALUES OF A PAIRED T-TEST BETWEEN SEGMENTATION RESULTS OF MODELS TRAINED WITH/WITHOUT CONFIDENCE MAPS.

Age (in month)	BCP					Philips
	18	12	9	6	3	6
GM	0.0371	0.2324	0.1934	0.1055	0.0027	0.0119
WM	0.2324	0.0330	0.0488	0.0417	0.0547	0.0020

Comment 1.7. “Finally, defining “loss encourages the segmentation model to pay more attention to these reliable labels, thus helping avoid anatomical errors” lacks justification as most likely “reliable labels” are inside a tissue region and the boundaries (i.e., the area of importance in a segmentation) most likely will be excluded from training as they are defined by ambiguous tissue distribution given the poor MRI contrast.”

Response: We agree with the reviewer that automatically-generated labels inside a tissue region are more reliable than these in the boundaries. Figure S12 shows the original T1w images from 6-month-old subjects, corresponding automated segmentations, confidence maps, and confidence maps after exclusion (by setting the confidence value as 0 if it is smaller than 0.5), from top to bottom. It can be seen that most of labels inside a tissue are reliable while labels in the boundaries are less reliable, e.g., the confidence value for the boundary voxel A in subject #1 is very low (0.35). However, it does not mean every label in the boundaries is unreliable, e.g., the confidence value for the boundary voxel B in subject #1 is very high (0.98), which will provide guidance for training. Moreover, we extract training samples from a number of testing subjects. Even the confidence value at a specific boundary in one testing subject could be low (e.g., the voxel A in subject #1), its corresponding confidence values in other testing subjects are possibly high (e.g., the voxel A’ in subject #2 with confidence value 0.95), thus providing guidance for training.

Fig. S12. The rows from top to bottom show, respectively, the original T1w images from 6-month-old subjects, corresponding automated segmentations, confidence maps, and confidence maps after exclusion (by setting the confidence value as 0 if it is smaller than 0.5). Confidence values for voxels A, B and A’ are 0.35, 0.98, and 0.95 respectively. Automatically-generated labels in the boundaries are still helpful for training, e.g., voxels B and A’.

In the revised manuscript, we have added Fig. S12 and the corresponding content in new Section V of the *Supplementary Materials*, and provided guidance in Section IV-B: “A justification for generating training samples in boundaries can be found in Section V of the *Supplementary Materials*”.

V. JUSTIFICATION OF GENERATING TRAINING SAMPLES IN BOUNDARIES

“In the proposed self-supervised learning strategy, automatically-generated labels inside a tissue region are more reliable than these in the boundaries. Figure S12 shows the original T1w images from 6-month-old subjects, corresponding automated segmentations, confidence maps, and confidence maps after exclusion (by setting the confidence value as 0 if it is smaller than 0.5), from top to bottom. It can be seen that most of labels inside a tissue are reliable while labels in the boundaries are less reliable, e.g., the confidence value for the boundary voxel A in subject #1 is very low (0.35). However, it does not mean every label in the boundaries is unreliable, e.g., the confidence value for the boundary voxel B in subject #1 is very high (0.98), which will provide guidance for training. Moreover, we extract training samples from a number of testing subjects. Even the confidence value at a specific boundary in one testing subject could be low (e.g., the voxel A in subject #1), its corresponding confidence values in other testing subjects are possibly high (e.g., the voxel A' in subject #2 with confidence value 0.95), thus providing guidance for training.”

Comment 2. *“The experiments are impressive on one side (the high accuracy of the proposed approach) but also misleading.”*

Comment 2.1. *“First, it is unclear why the approach was not tested on any 24 months old scans of BCP, which could have been done via cross-validation.”*

Response: Thanks for the comment. Compared with 18-, 12-, ..., 0-month-old subjects with low tissue contrast, **tissue segmentation of 24-months-old subjects is less challenging** due to their high tissue contrast. Therefore, we did not include the results on 24-months-old subjects in Table II in the original manuscript.

As suggested, we have now included the validation on 24-months-old subjects in Section VI-A in the revised manuscript. Specifically, we performed a 2-fold cross-validation with 10 repetitions to evaluate the segmentation performance, as pasted below:

VI-A. Validation in the Source Domain (24-month-old Subjects)

“In Table II and Section V-C, we have made quantitative validation on target domains from different time-points and sites. In the following, we will test the proposed method in the source domain via a 2-fold cross-validation with 10 repetitions. Specifically, in each repetition, eighteen 24-month-old subjects are randomly partitioned into two folds (9 subjects for each fold). Each fold is alternatively used as training subjects, while the other fold is treated as testing subjects. For comparison, we only choose the ADU-Net since it achieves better performance than volBrain, Infant FreeSurfer, and ASD-Net, based on Table II and Section V-C. Besides 9 subjects that are used for training, our method can also utilize the automatically-generated reliable samples from unlabeled testing subjects for training. Finally, the Dice ratios in 10 repetitions (% , Mean±Standard Deviation) achieved by the ADU-Net are 91.50±1.26, 90.92±1.19, 92.55±0.94 for CSF, GM and WM, respectively. In comparison, the proposed SSL method achieves slightly improved performance, i.e., 92.44±1.25, 91.39±1.10, and 93.09±1.73 for CSF, GM and WM, respectively. This experiment demonstrates that these automatically-generated reliable samples can also boost the segmentation performance in the source domain.”

Comment 2.2. *“As the proposed method is specifically designed to transfer segmentation results from 24 months to younger ages, the baselines listed in Table II do not seem to be a fair comparison.”*

Response: Thanks for pointing out this limitation. One of baseline methods, i.e., ASD-Net method, is trained using the similar strategy by transferring segmentation results from 24 months to younger months. Different from the proposed SSL method using confidence maps to select reliable labels and as weights (w) into the proposed spatially-weighted cross-entropy loss, the ASD-Net method considers all testing subjects in target domains as training samples and uses a conventional cross-entropy loss to train a domain-specific

segmentation model for each target domain. In the revised manuscript, we have clarified the training strategy of ASD-Net in Section V-A:

“For a fair comparison, the competing methods and our SSL share the same training subjects and testing subjects. Furthermore, we performed the same gradual label propagation strategy (i.e., 24->18->12->9->6->0~3) for ASD-Net method, which is detailed in Section IV-C. Specifically, ASD-Net method considers all testing subjects in target domains as training samples and uses a conventional cross-entropy loss to train a domain-specific segmentation model for each target domain”.

Comment 2.3. *“Another problem of Table II is that each statistic is probably based on the normal distribution over 5 accuracy scores (not specified if the numbers presented are mean and standard deviation) but the significance across approaches is based on a non-parametric test. Thus, for example, the significant differences between Proposed and ADU-net for the DICE-CSF score at 9 months is confusing given the high overlap between the distribution. However, making any inference based on 5 samples is questionable to start with. Similar comments apply to the remainder of the experimental section.”*

Response: Thanks for the instructive suggestion and valuable comment. We have revised the manuscript in the following several aspects:

(1) **Number of testing subjects for quantitative analysis:** In the revised manuscript, we have added 25 new testing subjects with manual labels from BCP dataset. Therefore, there are a total of 50 testing subjects with manual annotations for quantitative comparison in the revised manuscript. Please refer to Comment #1.3 of Reviewer 1 for more details on manual annotation.

(2) **Meaning of validation results:** In the revised manuscript, we specify the results in terms of “Mean \pm Standard Deviation” in the caption of Table II: *“Dice ratio and 95th-percentile Hausdorff distance (HD95) of cerebellum segmentation results on 50 testing subjects at 18, 12, 9, 6, and ≤ 3 months of age from BCP (Mean \pm Standard Deviation)”*. Please refer to our response to Comment #1.3 of Reviewer 1 for the updated Table II.

(3) **Method to calculate the statistical difference:** Regarding to the method of calculating p -values, we first used Jarque-Bera test to verify whether the accuracy scores come from a normal distribution or not. Although we have spent 300+ hours performing manual annotations to increase the number of testing subjects from original 5 to 10 for each month domain, the distribution of accuracy scores still does not fit a normal distribution. Therefore, a non-parametric test (Wilcoxon signed-rank test) is more appropriate to evaluate the statistical difference. The corresponding results have been updated in the Table II (please refer to Comment #1.3 of Reviewer 1).

(4) **DICE-CSF Results of 9-month-old subjects:** Sorry for our previous mistake on the significance test on the DICE-CSF. In the revised manuscript, we have validated on more testing subjects for statistical significance test, and we did not find any significant difference between ADU-Net and our proposed method.

Comment 3. *“In summary, the authors of this article are highly talented researchers that are trying to tackle a very difficult and important problem. However, the manuscript is not mature enough to be published in a high impact journal such as Nature Communications.”*

Response: We deeply appreciate the reviewer for the positive comments and the thoughtful suggestions, which help greatly improve the entire work. Accordingly, in the revised version, we have made the following major changes to thoroughly enhance this work:

- (1) We have redrawn Fig. 1 by following “global-to-local” and “coarse-to-fine” rules.
- (2) For a more intensive validation, we have spent over 300 h performing manual annotations to increase the number of testing subjects from original 25 to 50. Accordingly, we have updated Table II, Fig. 5, Table SI and Fig. S9, based on total 50 testing subjects with manual annotations.
- (3) We have included an infant cerebellum manual annotation protocol in Section I of Supplementary Materials, and added representative annotations for infant subjects at 0, 3, 6, 9, 12, and 18 months old in new Fig. S13.
- (4) We have provided a justification for generating training samples in boundaries in new Section V and new Fig. S12 of the Supplementary Materials.
- (5) We have performed 2-fold cross-validation with 10 repetitions to evaluate the segmentation model and confidence model in the source domain in Section VI-A and Fig. S8, respectively.
- (6) We have added more details on manual performers, accuracy metrics, and experiment setting in Section III, and Section V-A, e.g., the proposed gradual label propagation strategy was also used for competing methods for a fair comparison.
- (7) To focus on the cerebellum, we have moved the additional experiment on cerebrums (e.g., the original section “Application on Cross-site Infant Cerebrum Segmentation”) to the Supplementary Materials.

Comment 4: “*Minor points:*”

Comment 4.1. “*It probably would be good to mention in the paper that the authors are organizers of the iSeg challenges that are referred throughout.*”

Response: As suggested, in the revised manuscript, we have clearly indicated that we are the organizers in Section I and Section III of the *Supplementary Materials*, as pasted below:

Section I: “*As reported in a 6-month infant cerebrum segmentation challenge that we organized [13], i.e., iSeg-2019 (<http://iseg2019.web.unc.edu>), a model trained on a specific-site dataset usually performs well on testing subjects from the same site, but poorly on subjects from other sites with different imaging protocols/scanners.*”

Section III of the *Supplementary Materials*: “*Our framework is general and can be applied to other tasks, especially for those with multi-site datasets. Herein, we further validate the proposed SSL method for the infant cerebrum segmentation task in our organized iSeg-2019 challenge.*”

Comment 4.2. “*The article should be edited by a native speaker to avoid grammatical errors, difficult to understand sentences (such as the last one in Section II), and uncommon use of terminology such as cohort for data set.*”

Response: We have asked one native speaker to polish the manuscript.

Reviewer #2 (Remarks to the Author):

“While substantial advances have been made in precision segmentation and quantification of the cerebrum in early human development and developmental disorders, similar high quality segmentation of the cerebellum in infants to toddlers has not yet occurred. In ‘Self-Supervised Learning with Application for Infant Cerebellum Segmentation and Analysis’, Sun and colleagues propose a self-supervised learning (SSL) framework for cerebellum segmentation across the first two years postnatal. They used MRI scans from 333 subjects across these ages and compared SSL accuracy with other competing segmentation methods. Among the 333 subjects were 22 “ASD” and 73 “normal control” males to test performance of SSL to test whether SSL may be useful when applied to neurodevelopmental disorders. They used manual anatomical labelling of 5 subjects at each early age-point as testing subjects for validation of findings with “unlabeled” subject scans.

They successfully addressed a key the domain shift issue, a major challenge in the field. SSL outperforms other methods in across scanner platforms comparisons. The authors successfully used a “gradual label propagation strategy” starting with manually labelled 24 month old, then unlabeled 18 month old to develop accurate labelling of the most challenging youngest 12, 9 and 6 month old cerebellum images, showing very good to excellent performance especially at 18, 12, 9 and 6 months. This approach is superior to a direct propagation. They showed their method outperforms other deep-learning based segmentation methods. Examination of their figures and data shows impressive quality by SSL. They provide cerebellum gray and white matter growth curves across the first two years of postnatal life. They also applied the proposed method to segment the cerebellum of ASD toddlers and found that ASD boys have larger gray and white matter volume in cerebellum compared to ASD girls at 1 and 2 years old. The study and results are quite important.”

“I think this work is important to publish once the authors address several comments and concerns:”

Response: Thanks very much for the positive comment, which really encourages us.

Major Comment 1. *“The shape of the confidence map seems don’t match with the shape of the Automated segmentation or Manual label (Fig. 2), especially in the first and fourth rows. Can the authors explain?”*

Response: Thanks for pointing this out. In Fig. 2, the confidence maps are in 2D views, while the automated segmentations and manual labels are in 3D views. This is why it seems like there is some mismatch between the confidence map and segmentations. To avoid the confusion, in the revised manuscript, we have updated Fig. 2 to make them as consistent as possible, as pasted below.

Fig. 2. Confidence maps of automated segmentation results at different timepoints. The first column shows the confidence maps, the second column shows the segmentation results and the last column shows the corresponding manual labels. Note that the darker color in confidence maps means less confidence score, and vice versa. Red dashed circles indicate unreasonable results in the automated segmentations (e.g., “handle” and “hole”), which are reflected in dark color in the confidence map

Major Comment 2. *“How many random partitions are used in 2-fold cross-validation? If it is from one, please evaluate the performance from more partitions.”*

Response: Thanks for the instructive comment and suggestion. In the original manuscript, we performed 2-fold cross-validation by using one time random partition. As suggested, in the revised manuscript, we have performed 2-fold cross-validation with 10 repetitions to evaluate the performance in the source domain (i.e., 24-month-old infants). Please refer to Comment #2.1 of Reviewer 1 for more details.

Major Comment 3. *“The authors mentioned that a set of reliable training samples was generated for target domain (e.g., infants younger than 24 month) segmentation models. Please specify how many training samples were generated for each target domain model, and plot the stability index of those training samples and automated segmentation.”*

Response: Thanks for the valuable suggestion. In the proposed self-supervised training strategy, we utilize confidence maps to generate reliable training samples for each target domain. Herein, we investigate the performance in terms of the number of training samples by taking the segmentation of 18-month-old subjects for example, through a cross-validation. Basically, different numbers of samples from each of 10 unlabeled testing subjects are extracted and used to train a segmentation model, which is then applied to another 10 testing subjects (with manual labels) for a quantitative evaluation.

Specifically, we randomly selected 250/500/750/1,000/1,250/1,500 training samples from each of 10 unlabeled testing subjects, with the corresponding Dice ratio results for GM and WM shown in Fig. S11.

As expected, as the training sample number increases, the Dice ratios gradually increase. Considering accuracy and training complexity, we recommend extracting 1,000 training samples from each subject, as we mentioned in Section IV-C: “To train a segmentation model, we randomly extracted 1,000 MR image patches (size: $32 \times 32 \times 32$) from each subject, where the T1w and T2w images were treated as two channels in the proposed network. Experimental results on the influence of the number of training samples are reported in Section IV of the Supplementary Materials”.

Fig. S11. Results of Dice ratio in segmenting GM and WM with different numbers of training samples from each testing subject in the proposed self-supervised training strategy.

In the revised manuscript, we have added a new Section IV and a new Fig. S11 in the *Supplementary Materials*:

IV. INFLUENCE OF THE NUMBER OF TRAINING SAMPLES

“In the proposed self-supervised training strategy, we utilize confidence maps to generate a set of reliable training samples for each target domain. Herein, we investigate the performance in terms of the number of training samples by taking the segmentation of 18-month-old subjects for example, through a cross-validation. Basically, different numbers of samples from each of 10 unlabeled testing subjects are extracted and used to train a segmentation model, which is then applied to another 10 testing subjects (with manual labels) for a quantitative evaluation. Specifically, we randomly selected 250/500/750/1,000/1,250/1,500 training samples from each of 10 unlabeled testing subjects, with the corresponding Dice ratio results for GM and WM shown in Fig. S11. As expected, as the training sample number increases, the Dice ratios gradually increase. Considering accuracy and training complexity, we recommend extracting 1,000 training samples from each subject.”

Major Comment 4. *“Please provide the rationale of choosing Dice ratio and HD95 over others as evaluation metrics. What do these two metrics measure?”*

Response: Thanks for the comment. In this work, we use the Dice ratio and 95th percentile Hausdorff Distance (HD95) as metrics to evaluate the accuracy. Dice ratio is commonly used in segmentation accuracy assessment, which is volume-based calculation. Hausdorff distance (HD) is surface-based calculation, by measuring the distance between the estimated surface and manual surface, and HD95 is used to avoid the outliers as opposed to standard HD. Higher Dice ratio and lower HD95 values indicate better segmentation results. These details have been included in Section V-A in the revised manuscript.

Major Comment 5. *“Table II shows that WM segmentation accuracy is higher than GM, can authors explain and discuss about it?”*

Response: Thanks for the careful review. For Siemens scans, to further verify the superior WM accuracy in comparison of GM accuracy, we included 25 new testing subjects and still draw the same conclusion. The possible reasons are twofold. First, as GM is a narrow sheet between CSF and WM, we have to find both CSF/GM boundaries and GM/WM boundaries to identify GM, which is more challenging than identifying WM that only involves GM/WM boundaries. Second, tissue contrast between CSF and GM is much lower than that between GM and WM (as also confirmed from the much lower CSF accuracy). Therefore, finding CSF/GM boundaries is much more difficult than finding GM/WM boundaries, resulting in low accuracy of GM segmentation.

We have included these discussions as a new Section VI-D in the revised manuscript:

VI-D. WM Is Easier to Identify than GM?

“It can be observed from Table II and Fig. 5 that the Dice ratio of WM achieved by the proposed method is typically higher than that of GM. Similar phenomenon can be also observed from HD95. The possible reasons are twofold. First, as GM is a narrow sheet between CSF and WM, we have to find both CSF/GM boundaries and GM/WM boundaries to identify GM, which is more challenging than identifying WM that only involves GM/WM boundaries. Second, tissue contrast between CSF and GM is much lower than that between GM and WM (as also confirmed from the much lower CSF accuracy). Therefore, finding CSF/GM boundaries is much more difficult than finding GM/WM boundaries, resulting in low accuracy of GM segmentation.”

Major Comment 6. *“Can authors provide time and memory complexity of the methods listed in Table II?”*

Response: Thanks for the good suggestion. We have listed the time and memory complexity for methods in a new Table RI, as pasted below.

Table RI. Time and memory complexity during the testing stage for methods listed in Table II.

Method	Time complexity	Memory complexity	Note
volBrain	30 min	Not available	Submit to website (https://www.volbrain.upv.es/)
Infant FreeSurfer	90 min	6 GB	Processor: 64 x AMD EPYC 7313 16-Core Processor OS: RedHat 7 x86_64
Multi-atlas-based method	30 min	10 GB	Processor: 64 x AMD EPYC 7313 16-Core Processor OS: RedHat 7 x86_64
ASD-Net	5 min	985 MB	Processor: Intel® Core™ i9-8950HK CPU @ 2.90GHz × 12 GPU: NVIDIA GeForce RTX 2080 OS: Ubuntu 18.04.5 LTS
ADU-Net	5 min	1860 MB	Processor: GNU/Linux 5.13.0-44-generic x86_64 GPU: NVIDIA RTX A5000 OS: Ubuntu 20.04.4 LTS
Proposed	5 min	1860 MB	Processor: GNU/Linux 5.13.0-44-generic x86_64 GPU: NVIDIA RTX A5000 OS: Ubuntu 20.04.4 LTS

Major Comment 7. *“Table II shows that WM accuracy was consistently higher than GM accuracy in BCP dataset from the Siemens scanner, while Fig. 5 shows the opposite direction for images collected from the Philips scanner. Can authors discuss about it?”*

Response: Thanks for the valuable comments. For Philips scans, due to our carelessness, the heights of y-axes in two sub-figures are not the same, making the GM misleadingly look higher than WM. In fact, for the proposed method, the accuracy for WM (92.13%) is still higher than that for GM (90.91%). Accordingly, we have updated Fig. 5b with the same heights for y-axes, as pasted below. We have discussed why WM is easier to identify than GM as a new Section VI-D in the revised manuscript (please refer to Major Comment #5 of Reviewer 2).

Fig. 5b. Comparison of Dice ratios of GM and WM segmentation results between 10 Siemens testing subjects (i.e., BCP data) and 5 Philips testing subjects at 6 months of age, obtained by the five methods.

Major Comment 8. *“Please provide more information on these longitudinal ASD and normal control samples (e.g., ADOS scores, psychometric scores, age at diagnosis and diagnosis criteria, ethnicity, exclusion criterion, site and scanner information). The lack of ASD and NC information makes this unpublishable; if such information is not available, then I recommend removing this from the main text and putting this ASD vs NC into supplemental information. If the images were collected from multiple sites and scanners, it will be necessary to check whether site and scanner are confounded with the group difference results in Fig. 6 c-d. It will also be interesting to see whether cerebellar differences between ASD and NC at 12 months are present with and without taking total brain size.”*

Response: Thanks for the valuable comments and instructive suggestions.

(1) **Information of NDAR data:** Necessary information about NDAR data (i.e., the longitudinal ASD and normal control samples) is available from the following published article [1][2], including diagnosis criteria, ethnicity, data collection sites, MRI acquisition information and data exclusion criteria during collection. We pasted these information below for reference. In the revised manuscript, we have added the guidance information in Section III:

“The diagnosis of autism was made using the DSM-IV-TR criteria [34] at 24 or 36 months old. Twenty-two met clinical criteria for autism while 73 were included as normal control (NC). More details on the studied subjects, including diagnosis criteria, ethnicity, data collection sites, MRI acquisition information and data exclusion criteria during collection, can be found in [32], [33].”

- a. **Diagnosis criteria:** DSM-IV-TR criteria [3] at 24 months or 36 months old of age.
- b. **Ethnicity:** White (~87%), non-white (~12%), not reported (~1%).
- c. **Behavioral assessment:** Infants were assessed directly at ages 6, 12 and 24 months, and a phone interview was conducted with parents at 18 months to assess infant development. Direct assessments include brain MRI scans in addition to a battery of behavioral and developmental tests. The assessment battery for the visit at 6 months included the Mullen Scales of Early Learning (Mullen), the Vineland Adaptive Behavior Scales-II, the Autism Observation Scale for Infants, various questionnaires examining behavior, temperament, and family characteristics, and a medical record review.

- d. **Sites:** A number of quality control procedures were employed to assess scanner stability and reliability across sites, time, and procedures. A ‘LEGO’ phantom was scanned monthly at each location and analyzed for image quality and to quantitatively address site specific regional distortions. Two adult subjects (aka ‘human phantoms’) were scanned once per year per scanner (twice in year 1). Phantom data was evaluated for scanner stability across sites and time. Results indicated excellent stability across sites, with covariates of variation for intracranial volume below 1%, and with intraclass correlations for intracranial volume at 0.98 for inter-site and 0.99 for intra-site reliability. Finally, all scans were blindly reviewed for image quality by a single rater (D. Louis Collins, McGill), and again rated by a single reviewer (Rachel Gimpel Smith, UNC) in a pre-processing stage prior to image analysis.
- e. **MRI acquisition:** All the brain MRI scans were completed at each of the clinical sites on a 3T Siemens Tim Trio scanner with a 12-channel head coil. All scans were completed while infants were naturally sleeping. Specific structured preparation was completed by families at home prior to the scanning, including conditioning to the scanner sounds on a CD played to infants while sleeping. The imaging protocol was designed to maximize tissue contrast for volumetric analysis across three timepoints (ages 6, 12, 24 months). The protocol included (a) a localizer scan, (b) 3D T1 MPRAGE: TR=2400ms, TE=3.16ms, 160 sagittal slices, FOV=256, voxel size = 1mm³, (c) 3D T2 FSE TR=3200ms, TE=499ms, 160 sagittal slices, FOV=256, voxel size = 1mm³, and (d) a 25 direction DTI: TR=12800ms, TE=102ms, slice thickness = 2mm isotropic, variable b value = maximum of 1000s/mm², FOV=190.
- f. **Data exclusion criteria during collection:** According to Hazlett, H. et al. paper [1], for theNDAR dataset, subjects were enrolled as high familial risk for ASD (HR) if they had an older sibling with a clinical diagnosis of ASD confirmed with the Autism Diagnostic Interview-Revised(ADI-R). Subjects were enrolled in the low familial risk (LR) group if they had an older sibling without evidence of ASD and no family history of a first or second-degree relative with ASD. Exclusion criteria for both groups included the following: (a) diagnosis or physical signs strongly suggestive of a genetic condition or syndrome (for example, fragile X syndrome) reported to be associated with ASDs, (b) a significant medical or neurological condition affecting growth, development or cognition (for example, CNS infection, seizure disorder, congenital heart disease), (c) sensory impairment such as vision or hearing loss, (d) low birth weight (< 2,000 g) or prematurity (< 36 weeks gestation), (e) possible perinatal brain injury from exposure to in utero exogenous compounds reported to likely affect the brain adversely in at least some individuals (for example, alcohol, selected prescription medications), (f) non-English speaking families, (g) contraindication for MRI (for example, metal implants), (h) adopted subjects, and (i) a family history of intellectual disability, psychosis, schizophrenia or bipolar disorder in a first-degree relative. The sample for this analysis included all children with longitudinal imaging data processed until 31 August 2015.

(2) **Site and scanner:** A number of quality control procedures were employed to assess scanner stability and reliability across sites, time, and procedures. A ‘LEGO’ phantom was scanned monthly at each location and analyzed for image quality and to quantitatively address site specific regional distortions. Two adult subjects (aka ‘human phantoms’) were scanned once per year per scanner (twice in year 1). Phantom data was evaluated for scanner stability across sites and time. Results indicated excellent stability across sites, with covariates of variation for intracranial volume below 1%, and with intraclass correlations for intracranial volume at 0.98 for inter-site and 0.99 for intra-site reliability. Finally, all scans were blindly reviewed for image quality by a single rater (D. Louis Collins, McGill), and again rated by a single reviewer (Rachel Gimpel Smith, UNC) in a pre-processing stage prior to image analysis.

(3) **Cerebellar differences between ASD and NC at 12 months:** As suggested, we calculate the cerebellum volumes and normalized cerebellum volumes in terms of total brain volume (TBV) between NC and autistic subjects at 12 months, as shown in Fig. 7. We can see that at 12 months of age, the cerebellum volumes of autistic subjects are larger than that of NC subjects with significant difference (p -value = 0.0286), while there is no significant difference (p -value = 0.4641) with the normalized cerebellum volumes in terms of TBV. Furthermore, we also report the cerebellum volumes and normalized cerebellum volumes in terms of TBV between NC and autistic subjects at 6 months and 24 months. From Fig. 7, as infants grow up, the cerebellum volumes between NC and autistic infants gradually become significantly different (6 months: p -value = 0.2798, 12 months: p -value = 0.0286, 24 months: p -value = 0.0429), whereas the normalized cerebellar volume has no significant difference (6 months: p -value = 0.4246, 12 months: p -value = 0.4641, 24 months: p -value = 0.4831).

Fig. 7. **a.** Cerebellum volumes for NC and autistic infants at 6/12/24 months. **b.** Normalized cerebellum volumes in terms of total brain volume (TBV) for NC and autistic infants at 6/12/24 months.

In the revised manuscript, we have added new Fig. 7 in Section V-D, as well as the corresponding content:

“We further calculate cerebellum volumes and normalized cerebellum volumes in terms of total brain volume (TBV) between NC and autistic subjects at 6 months, 12 months and 24 months, as shown in Fig. 7. From Fig. 7, as infants grow up, the cerebellum volumes between NC and autistic infants gradually become significantly different (6 months: p -value = 0.2798, 12 months: p -value = 0.0286, 24 months: p -value = 0.0429), whereas the normalized cerebellar volume has no significant difference (6 months: p -value = 0.4246, 12 months: p -value = 0.4641, 24 months: p -value = 0.4831).”

[1] H. C. Hazlett, H. Gu, B. C. Munsell, S. H. Kim, M. Styner, J. J. Wolff, J. T. Ellison, M. R. Swanson, H. Zhu, K. N. Botteron, D. L. Collins, J. N. Constantino, S. R. Dager, A. M. Estes, A. C. Evans, V. S. Fonov, G. Gerig, P. Kostopoulos, R. C. McKinstry, J. Pandey, S. Paterson, J. R. Pruett, R. T. Schultz, D. W. Shaw, L. Zwaigenbaum, J. Piven, and T. I. Network, “Early brain development in infants at high risk for autism spectrum disorder,” *Nature*, vol. 542, p. 348–351, 2017.

[2] H. C. Hazlett, H. Gu, K. N. B. S. D. Robert C. McKinstry, Dennis W.W. Shaw, M. Styner, C. Vachet, G. Gerig, R. T. Sarah Paterson, A. M. Estes, A. C. Evans, and J. Piven, “Brain volume findings in 6-month-old infants at high familial risk for autism,” *The American journal of psychiatry*, vol. 169, no. 6, pp. 601–608, 2012.

[3] American, Psychiatric, and Association, *Diagnostic and Statistical Manual of Mental Disorders (DSM-IV-TR)*: American Psychiatric Association, 2000.

Major Comment 9. *“The literature that motivates measuring the cerebellum in ASD is needs to be better represented.”*

Response: Thanks for this valuable suggestion. In the revised manuscript, we have added the motivation in Section V-D:

“Diagnosis of autism at an early age is highly desirable, as early intervention can result in long-term positive effects on symptoms. The cerebellum is one of the most consistently reported structures to be affected by autism, as reported in previous studies on subjects with 4+ years old [49], [50]. However, our knowledge on cerebellar growth of the

autistic brain in the early postnatal stages remains extremely limited. Therefore, in this work, we further investigate whether the cerebellar growth trajectories are different between NC subjects and autistic subjects by leveraging the third dataset with 95 longitudinal subjects scanned at 3 time-points: 6 months, 12 months, and 24 months of age. ”

Minor Comment 1. *“In introduction, the authors mentioned that the anatomical errors in the segmentation results are shown in the left bottom of Fig. 1c, but I did not see them in Fig. 1c.”*

Response: For a better interpretation of anatomical errors, we have clearly indicated the anatomical errors in the second column of Fig. 2 in the revised manuscript:

“Third, the arbor vitae is a complete and folded tree-like appearance; however, due to extremely low tissue contrast and severe partial volume effect, there are often anatomical errors (“hole” and “handle”) in the segmentation results, as shown in the second column of Fig. 2”.

Minor Comment 2. *“Are structural scans in the UNC/UMN BCP collected from only normal controls? Do they contain longitudinal scans?”*

Response: Subjects from the UNC/UMN BCP project are all normal controls [1], and the detailed data exclusion criteria is pasted below for reference. The UNC/UMN BCP project does contain longitudinal scans, but we only include cross-sectional scans in this work. We have provided this information in Section III of the manuscript:

“(1) The first dataset consists of 276 cross-sectional subjects (with 18 labeled images used for training, 50 labeled images and 208 unlabeled images used for validation) from the UNC/UMN Baby Connectome Project (BCP) [31], where images were acquired at $\leq 3, 6, 9, 12, 18,$ and 24 months of age using a Siemens Prisma scanner. Subjects from BCP are all normal controls, and the data exclusion criteria during collection can be found in [31]. ”

Data exclusion criteria during collection in BCP: According to Howell, B.R. et al. paper [1], for the BCP dataset, subjects are excluded if they were born prior to 37 weeks gestation, had a birth weight lower than 2,000 grams, or if they had any major delivery complications. Major delivery complications may include neonatal hypoxia or neonatal illness requiring a greater than two day NICU stay. They are also excluded if they: (1) are adopted, (2) have a first degree relative with autism, intellectual disability, schizophrenia, or bipolar disorder, (3) have any significant medical and/or genetic conditions affecting growth, development, or cognition, or (4) have any contraindication to MRI. Additional exclusion criteria include major pre-and/or perinatal issues including: maternal pre-eclampsia, placental abruption, maternal HIV status, and maternal alcohol or illicit drug use during pregnancy. Finally, children are excluded from the study if their caregivers are unable to communicate in English at a level to provide informed consent.

[1] B. Howell, M. Styner, W. Gao, P. Yap, L. Wang, K. Baluyot, E. Yacoub, G. Chen, T. Potts, A. Salzwedel, G. Li, J. Gilmore, J. Piven, J. Smith, D. Shen, K. Ugurbil, H. Zhu, W. Lin, and J. Elison, “The UNC/UMN baby connectome project (BCP): An overview of the study design and protocol development,” *NeuroImage*, vol. 185, pp. 891–905, Jan. 2019.

Reviewers' comments:

Reviewer #1 (Remarks to the Author):

“In the last review I wrote that the article misses rigor and focus, which is sadly enough still the case this time around. Instead of responding point-by-point to the response by the reviewer, I will just review the abstract in hope that this is informative for the authors.”

Comment 1. *“The abstract states “we propose a novel self-supervised learning framework for infant cerebellum segmentation” - however the abstract fails to clarify what is so novel. Even more disappointing is that even the intro fails to do so as concepts like using the segmentation of one time point to guide others have been explored in medical image analysis for the last 20 years.”*

Response: Thanks for pointing out this confusion. We acknowledge that over the last two decades, numerous studies have been published that “using some kind of one time point to guide others”. However, our work introduces the first accurate cerebellum segmentation model for challenging infant subjects. We achieve this through a novel self-supervised learning approach, and we are excited to share this with the scientific community. By leveraging the proposed strategy, we can obtain a domain-dedicated segmentation model based on a set of automatically generated reliable training samples, even for a domain without manual annotations.

To avoid the confusion for the proposed method, we have revised the corresponding content in Abstract of the updated manuscript: *“In this paper, we propose an accurate self-supervised learning framework for infant cerebellum segmentation in the first two postnatal years”*.

Comment 2. *“Next, the abstract states “Experimental results on 358 subjects from three datasets demonstrate superior performance of our framework” The word performance has many meanings and thus requires further specification especially for the broad readership of NComm. One common use of the word is measuring the speed of the algorithm, which the experiments do not do. More disappointing is that the accuracy of the method is only quantitatively assessed on 56 cases, in which the algorithm is significantly better than other approaches for only some ages and tissues.”*

Response: Thanks for your helpful suggestion. The “performance” means accuracy (the “performance” has been changed as “accuracy” in the updated manuscript). Regarding the speed of our method, we have included a new Table SIV in the *Supplementary Materials* that compares the time and memory complexity of our method with other non-deep-learning methods, such as *volBrain*, *Infant FreeSurfer*, and multi-atlas-based methods. Our method is much faster than these other methods during the testing stage. We have added the corresponding content in Section V-B of the revised manuscript: *“In terms of speed, our method is faster than other non-deep-learning methods (volBrain, Infant FreeSurfer, and the multi-atlas-based method) according to Table SIV in the Supplementary Materials by comparing the time and memory complexity during the testing stage between different methods”*.

In addition, we have also included a new Table SII in the *Supplementary Materials*, which provides an overview of available cerebellum tissue segmentation methods for brain MRIs, along with the number of testing subjects for each method. Upon reviewing Table SII, we observed that only *Infant FreeSurfer* and ADU-Net have been quantitatively evaluated on 17 and 10 infant subjects, respectively. Notably, our study has quantitatively assessed the segmentation method on a larger cohort of 55 subjects, which is the largest sample size for infant cerebellum analysis to date. In Sections V-B and V-C of the main text, we have compared our proposed method with *Infant FreeSurfer* and ADU-Net, both quantitatively and qualitatively.

While our method may not outperform others for every metric, our results demonstrate that it outperforms other methods for 73% of all metrics (96/130), representing a significant improvement in the field.

In the revised manuscript, we have added the corresponding contents to compare the number of testing subjects in Section III: “It’s worth noting that a total of 55 testing subjects were used for quantitative evaluation in this study, combining the first and second datasets. This is currently the largest sample size used for infant cerebellum analysis, as indicated in Table SII of the Supplementary Materials, which provides a summary of available cerebellum tissue segmentation methods for brain MRIs”.

TABLE SIV
TIME AND MEMORY COMPLEXITY DURING THE TESTING STAGE FOR COMPETING AND THE PROPOSED METHODS.

Method	Time	Memory	Note
volBrain [17]	30 min	Not available	Submit to website (https://www.volbrain.upv.es)
Infant FreeSurfer [7]	90 min	6 GB	Processor: 64 x AMD EPYC 7313 16- Core Processor; OS: RedHat 7 x86_64
Multi-atlas-based method [18]	30 min	10 GB	Processor: 64 x AMD EPYC 7313 16- Core Processor; OS: RedHat 7 x86_64
ASD-Net [19]	5 min	0.96 GB	Processor: Intel® Core™ i9-8950HK CPU @ 2.90GHz × 12; GPU: NVIDIA GeForce RTX 2080; OS: Ubuntu 18.04.5 LTS
ADU-Net [16]	5 min	1.82 GB	Processor: GNU/Linux 5.13.044 generic x86_64; GPU: NVIDIA RTX A5000; OS: Ubuntu 20.04.4 LTS
Proposed	5 min	1.82 GB	Processor: GNU/Linux 5.13.044 generic x86_64; GPU: NVIDIA RTX A5000; OS: Ubuntu 20.04.4 LTS

TABLE SII
AVAILABLE CEREBELLUM TISSUE SEGMENTATION METHODS FOR BRAIN MRIS.

Methods	Quantitative evaluation with infant subjects	Number of testing subjects
volBrain [6]	×	-
Infant FreeSurfer [7]	✓	17
SegNet [8]	×	-
FastSurfer [9]	×	-
HighRes3DNet [10]	×	-
PICSL [11]	×	-
FreeSurfer [12]	×	-
SLANT [13]	×	-
AssemblyNet [14]	×	-
ADU-Net [15], [16]	✓	10
Proposed	✓	55

- [6] J. E. Romero, P. Coupé, R. Giraud, V.-T. Ta, V. Fonov, M. T. M. Park, M. M. Chakravarty, A. N. Voineskos, and J. V. Manjón, "CERES: A new cerebellum lobule segmentation method," *NeuroImage*, vol. 147, pp. 916–924, 2017.
- [7] L. Zöllei, J. E. Iglesias, Y. Ou, P. E. Grant, and B. Fischl, "Infant freesurfer: An automated segmentation and surface extraction pipeline for t1-weighted neuroimaging data of infants 0–2 years," *NeuroImage*, vol. 218, p. 116946, 2020.
- [8] V. Badrinarayanan, A. Kendall, and R. Cipolla, "Segnet: A deep convolutional encoder-decoder architecture for image segmentation," *IEEE Transactions on Pattern Analysis and Machine Intelligence*, vol. 39, no. 12, pp. 2481–2495, 2017.
- [9] L. Henschel, S. Conjeti, S. Estrada, K. Diers, B. Fischl, and M. Reuter, "Fastsurfer - a fast and accurate deep learning based neuroimaging pipeline," *NeuroImage*, vol. 219, p. 117012, 2020. [Online]. Available: <https://www.sciencedirect.com/science/article/pii/S1053811920304985>
- [10] W. Li, G. Wang, L. Fidon, S. Ourselin, M. J. Cardoso, and T. Vercauteren, "On the compactness, efficiency, and representation of 3d convolutional networks: Brain parcellation as a pretext task," in *Information Processing in Medical Imaging*, M. Niethammer, M. Styner, S. Aylward, H. Zhu, I. Oguz, P.-T. Yap, and D. Shen, Eds. Cham: Springer International Publishing, 2017, pp. 348–360.
- [11] H. Wang and P. Yushkevich, "Multi-atlas segmentation with joint label fusion and corrective learning—an open source implementation," *Frontiers in Neuroinformatics*, vol. 7, 2013. [Online]. Available: <https://www.frontiersin.org/articles/10.3389/fninf.2013.00027>
- [12] B. Fischl, "Freesurfer," *NeuroImage*, vol. 62, no. 2, pp. 774–781, 2012, 20 YEARS OF fMRI. [Online]. Available: <https://www.sciencedirect.com/science/article/pii/S1053811912000389>
- [13] Y. Huo, Z. Xu, Y. Xiong, K. Aboud, P. Parvathaneni, S. Bao, C. Bermudez, S. M. Resnick, L. E. Cutting, and B. A. Landman, "3d whole brain segmentation using spatially localized atlas network tiles," *NeuroImage*, vol. 194, pp. 105–119, 2019. [Online]. Available: <https://www.sciencedirect.com/science/article/pii/S1053811919302307>
- [14] P. Coupé, B. Mansencal, M. Clément, R. Giraud, B. Denis de Senneville, V.-T. Ta, V. Lepetit, and J. V. Manjón, "Assemblynet: A large ensemble of cnns for 3d whole brain mri segmentation," *NeuroImage*, vol. 219, p. 117026, 2020. [Online]. Available: <https://www.sciencedirect.com/science/article/pii/S1053811920305127>
- [15] J. Chen, H. Zhang, D. Nie, L. Wang, G. Li, W. Lin, and D. Shen, "Automatic accurate infant cerebellar tissue segmentation with densely connected convolutional network," *Machine learning in medical imaging. MLMI*, vol. 11046, pp. 233–240, 2018.
- [16] L. Wang, G. Li, F. Shi, X. Cao, C. Lian, D. Nie, M. Liu, H. Zhang, Z. Wu, W. Lin, and D. Shen, "Volume-based analysis of 6-month-old infant brain MRI for autism biomarker identification and early diagnosis," in *MICCAI*, vol. 11072, 2018, pp. 411–419.
- [17] J. V. Manjón and P. Coupé, "volBrain: An online MRI brain volumetry system," *Frontiers in Neuroinformatics*, vol. 10, p. 30, 2016.
- [18] L. Wang, K. C. Chen, F. Shi, S. Liao, G. Li, Y. Gao, S. G. Shen, J. Yan, P. K. M. Lee, B. Chow, N. X. Liu, J. J. Xia, and D. Shen, "Automated segmentation of cbct image using spiral ct atlases and convex optimization," in *MICCAI*. Berlin, Heidelberg: Springer Berlin Heidelberg, 2013, pp. 251–258.
- [19] D. Nie, Y. Gao, L. Wang, and D. Shen, "ASDNet: Attention based semi-supervised deep networks for medical image segmentation," in *Medical Image Computing and Computer Assisted Intervention – MICCAI 2018*, A. F. Frangi, J. A. Schnabel, C. Davatzikos, C. Alberola-López, and G. Fichtinger, Eds. Cham: Springer International Publishing, 2018, pp. 370–378.

Comment 3. "One interesting point is that "GM and WM volumes are larger in autistic males than controlled males". However, as also pointed out by Figure 7 b), that difference disappears once normalized for total brain volume. Thus, the difference might be due to group differences in overall body size, race, or acquisition but as this (demographic) information is not provided by the article or further discussed one cannot make any conclusions about the findings.

Thus I am coming to the conclusion regarding the final statement of the abstract, i.e., "Our method and findings may shed light on a better understanding of cerebellum structure and function, as well as the diagnosis and treatment of early neurodevelopmental disorders." that it does not shed light on these issues but requires further investigation into this difficult and important problem."

Response: For our group analysis, we utilized subjects who were scanned at the same age (i.e., longitudinally at 6, 12, and 24 months of age), of the same gender (i.e., all male subjects), and almost the same race (with White subjects representing 90.9% of the autism group and approximately 89.0% of the normal control group), all of whom were scanned using a Siemens 3T Tim Trio scanner. It is important to note that, from all the subjects available in NDAR, only 95 were scanned longitudinally at all three time points mentioned above. Of these 95 subjects, 22 met clinical criteria for autism, while the remaining 73 were included as normal controls. We included all of them in our study instead of selectively choosing based on specific criteria. We have added a new Section VI and two new tables (i.e., Table SV and Table SVI) to the *Supplemental Materials* in the revised manuscript. These tables provide more demographic information about the autistic and normal control groups, including ethnicity, behavioral assessment, site consistency, data exclusion criteria during collection, cerebellar tissue volumes, and related clinical measures in terms of Mullen and ADOS.

It is important to note that when normalized by total brain volume, a "difference disappears" result does not necessarily imply that there are no differences between two groups. To address this issue, we have included a new Table SIII in the *Supplemental Materials* of the revised manuscript. This table calculates the significant difference of cerebellum volumes, normalized cerebellum volumes and normalized cerebellar

GM and WM volumes in terms of the total brain volumes between 22 autistic subjects and 73 normal controls. Although there is no significant difference between the two groups for cerebellum volumes after normalization, the normalized cerebellar GM volumes have small effect size at 6 and 24 months (6 months: Cohen's $d = 0.2735$; 24 months: Cohen's $d = 0.2477$), indicating that the difference between groups is not so small as to be trivial. This difference needs to be further investigated when more subjects become available in the future. In the revised manuscript, we have updated the corresponding content in Section V-E:

“We further calculate cerebellum volumes, normalized cerebellum volumes in terms of TBV, and normalized cerebellar GM and WM volumes in terms of TBV between NC and autistic subjects at 6 months, 12 months and 24 months, as shown in Fig. 7. We found that as infants grow up, the difference of cerebellum volumes between NC and autistic infants gradually increases. Specifically, based on Wilcoxon rank sum test in Table SIII of the Supplementary Materials, the difference in cerebellum volumes is not significant at 6 months (p -value = 0.1647), but becomes significant at 12 months (p -value = 0.0162) and 24 months (p -value = 0.0265). However, the normalized cerebellar volumes in terms of TBV have no significant difference. In addition, for the normalized cerebellar GM and WM volumes as shown in Fig. 7c-d, we did not find any significant difference in these trends between the NC and autistic groups, but the normalized cerebellar GM volumes have small effect size at 6 and 24 months (6 months: Cohen's $d = 0.2735$; 24 months: Cohen's $d = 0.2477$), indicating that the difference between groups is not so small as to be trivial. This difference needs to be further investigated when more subjects are available in the future.”

In terms of the “final statement of the abstract”, we have clearly clarified the data limitation and indicated the corresponding findings need to be further proved in the revised Abstract: *“Our finding indicate that the first six months may be the most rapid and dynamic period, and gray matter (GM) may play a dominant role over white matter (WM) in the rapid growth of the cerebellum. We also find that both GM and WM volumes are larger in males than females, and that GM and WM volumes are larger in autistic males than controlled males. Although **these findings are limited by the number of subjects**, accurate segmentation results by our method on more subjects may shed light on a better understanding of cerebellum structure and function, and will advance insight into the neural and biological bases of developmental disorders.”*

All updated and newly added sections, figures and tables are pasted below for reference.

New Section VI of the Supplemental Materials: *“For reference, we provide the necessary subject screening information about the NDAR data [3]-[5]. Table SV and Table SVI list the cerebellar tissue volumes and related clinic measures for 22 autistic subjects and 73 normal control subjects from NDAR. The following information is available for each subject:*

- 1. **Behavioral assessment:** Infants were assessed at 6, 18, and 24 months using the Mullen Scales of Early Learning, the Vineland Adaptive Behavior Scales-II, the Autism Observation Scale for Infants, various questionnaires examining behavior, temperament, family characteristics, and a medical record review. These assessments are available in NIMH Data Archive (https://nda.nih.gov/edit_collection.html?id=19).*
- 2. **Site consistency:** As reported in [5], the infants were recruited, scanned, and accessed from four clinical data collection sites (University of North Carolina at Chapel Hill, University of Washington, Children's Hospital of Philadelphia, Washington University in St. Louis), a Data Coordinating Center at the Montreal Neurological Institute (McGill University), and two image processing sites (University of Utah and UNC). To ensure the stability of assess scanners and the reliability of brain MRI scans across different sites, time-points, and procedures, a number of quality control procedures were employed, including phantom data evaluation for site stability, and blind review for all scans. Results indicate excellent stability across sites, and two expert raters reviewed all scans to ensure the image quality.*
- 3. As reported in [5], **data exclusion criteria during collection** include: (a) genetic condition or syndrome associated with ASDs, (b) significant condition affecting development, (c) sensory impairment, (d) birth weight less than 2,000 g or prematurity less than 36 weeks gestation, (e) possible brain injury during the perinatal period, (f) non-English speaking families, (g) contraindication for MRI, (h) children who do not have a biological relationship with parents or siblings, and (i) first-degree relatives with intellectual disability, psychosis, schizophrenia, or bipolar disorder.*

Reviewer #2 (Remarks to the Author):

“This self-supervised learning method for segmenting gray and white matter in the 0 to 24 month old cerebellum is the most sophisticated of its type to date, and the effort is large. It is head and shoulders above previous efforts to perform precision measurements of gray and white matter in the cerebellum during early development.”

Response: Your comments are greatly appreciated, and we are glad to hear that you found our work valuable. We take your suggestions seriously and have worked to ensure that our research is conducted with the utmost rigor and integrity. Thank you for taking the time to provide us with such constructive comments.

“I have read the authors’ responses and manuscript revision and am largely satisfied with them with the following exception having to do with my previous questions about the autism sample and analyses.”

Comment 1. *“The sample is very small at only $n=22$ males and should be characterized by the authors as a small, preliminary sample intended to test the potential utility of the authors’ approach for detecting early age deviations of cerebellar anatomic development. $N=22$ does not represent the heterogeneity of ASD brain development, and results should be considered proof of principle and cannot be considered definitive.”*

Response: Thank you for your constructive suggestions. NDAR has a limited representation of infant subjects, with most subjects being 2+ years old. Nonetheless, we have found the largest sample to date of 22 autistic males who underwent longitudinal brain scans at 6, 12, and 24 months in the NDAR dataset. As suggested, we acknowledge the limitations of our study, particularly the small sample size of 22 male subjects with autism, which may limit the generalizability of our findings. Our results should be considered as a proof of principle and further investigation with a larger and more diverse sample is needed to confirm our findings. In the revised manuscript, we have emphasized this limitation to provide a clear understanding of the scope of our study:

(1) In **Abstract**, we have clarified the data limitation and indicated the corresponding findings need to be further proved: *“Our finding indicate that the first six months may be the most rapid and dynamic period, and gray matter (GM) may play a dominant role over white matter (WM) in the rapid growth of the cerebellum. We also find that both GM and WM volumes are larger in males than females, and that GM and WM volumes are larger in autistic males than controlled males. Although **these findings are limited by the number of subjects**, accurate segmentation results by our method on more subjects may shed light on a better understanding of cerebellum structure and function, and will advance insight into the neural and biological bases of developmental disorders.”*

(2) In **Section V-D**, we have noted that the growth trajectories of normal cerebellar development may suffer from limited data: *“Notably, the trajectory of infant cerebellum development may suffer from the limited number of studied subjects in this work, but it may be still worthy of reference for future work.”*

(3) In **Section V-E**, we have added the clarification that the cerebellar volume analysis of autistic subjects is a proof of principle rather than a definitive conclusion: *“In addition, for the normalized cerebellar GM and WM volumes as shown in Fig. 7c-d, we did not find any significant difference in these trends between the NC and autistic groups, but the normalized cerebellar GM volumes have small effect size at 6 and 24 months (6 months: Cohen’s $d = 0.2735$; 24 months: Cohen’s $d = 0.2477$), indicating that the difference between groups is not so small as to be trivial. **This difference needs to be further investigated when more subjects are available in the future. Please note that the above analysis is more like a proof of the principle that the more accurate segmentations at early ages will advance insight into the neural and biological bases. The corresponding findings are not considered to be definitive conclusions due to limited data.**”*

(4) In **Section VI-F**, we have further indicated the data limitation both for cross-sectional and longitudinal analysis: *“Third, the growth trajectory suffers from a limited number of cross-sectional/longitudinal subjects. As a result, the corresponding findings and conclusions need to be further validated when more subjects become available in the future.”*

(5) In **Section VII**, we have again indicated the corresponding findings need to be further proved: *“Importantly, we chart the trajectory of early cerebellum development in terms of different tissue types and find that GM plays a dominant role in the rapid growth of the cerebellum over WM. We further compare the development in terms of gender during the first two postnatal years and find that both GM and WM volumes are larger in males compared to females from birth to 24 months old. Additionally, we find that GM and WM volumes are larger in male autistic infants than in normal controls from 6 to 24 months old. **However, further studies are needed to confirm these findings when more subjects become available in the future.**”*

Comment 2. *“Despite the response to my question about the ASD subjects sample, it remains poorly clinically characterized. The main text does not say (1) where the 22 subjects came from (where was each child recruited and scanned?); (2) how these 22 ASD subjects were selected (was there bias in choosing these few cases?); what their symptom severity scores were (ADOS scores?); and what their language and cognitive scores were. This information is the least that should be included in the main text. Additional detail (such as the detail in the authors’ responses) can be placed in the Supplement.”*

Response: Thank you for your helpful suggestion to improve the integrity of our research. The infants in our study were recruited, scanned, and accessed from multiple clinical data collection sites and processing centers (University of North Carolina at Chapel Hill, University of Washington, Children’s Hospital of Philadelphia, Washington University in St. Louis), a Data Coordinating Center at the Montreal Neurological Institute (McGill University), and two image processing sites (University of Utah and UNC). Of the subjects available in NDAR, only 95 were longitudinally scanned at all three time-points (6, 12, and 24 months), and we included all of them in our analysis. Among these subjects, 22 met clinical criteria for autism while 73 were included as normal controls. We have added this information to Section III of the revised manuscript.

In response to your suggestion, we have also provided additional information about the autistic subjects, including the collection sites, diagnosis, and selection criteria. We have included this information in Section III of the revised manuscript and have added a new Section VI and two new tables (Table SV and Table SVI) in the *Supplemental Materials*. These tables provide detailed information about the behavioral assessments, site consistency, data exclusion criteria, cerebellar tissue volumes, ethnicity, and related clinical measures (such as Mullen and ADOS) for each subject used in our study. Please refer to Comment #3 of Reviewer 1 for the new Section VI, Table SV, and Table SVI of the *Supplemental Materials*.

Section III: *“The third dataset is from the National Database for Autism Research (NDAR) [31],[32] (<https://nda.nih.gov/about.html>) and includes 95 male subjects. This dataset was used to investigate whether there are differences in cerebellar growth trajectories between NC subjects and autistic subjects during the first two postnatal years. The infants were recruited, scanned, and accessed from four clinical data collection sites, a Data Coordinating Center, and two image processing sites (University of North Carolina at Chapel Hill, University of Washington, Children’s Hospital of Philadelphia, Washington University in St. Louis, McGill University, University of Utah and UNC). The data exclusion criteria during collection is available in Section VI of the Supplementary Materials. The diagnosis of autism was made using the DSM-IV-TR (Diagnostic and Statistical Manual of Mental Disorders, 4th Edition, Text Revision) criteria [33] at 24 or 36 months old. All images were acquired by a Siemens 3T Siemens Tim Trio scanner with a 12-channel head coil. Quality control procedures were employed to ensure image quality across different sites, times, and procedures, as described in Section VI of the Supplementary Materials. More information on the subjects studied [31], [32], including behavioral assessment and data exclusion criteria during collection, can be found in Section VI of the Supplementary Materials. Of all the subjects available in NDAR, only 95 were*

longitudinally scanned at all three time-points (6, 12, and 24 months of age), with 22 meeting clinical criteria for autism and 73 included as NCs. In the Supplementary Materials, Table SV and Table SVI list gender, race and related clinical measures (e.g., Mullen [34] and ADOS [35]) for each subject.”

Comment 3. *“The manuscript focuses on cerebellar gray and white growth trajectories which is excellent, but the statistical analyses of the ASD and control cerebellum data are not growth or change, but appear to be t-test at each study age. Also, the range of values shown in Figure 6 is quite large (which is expected) and the absolute volume differences are very small, and a $p < 0.05$ t-test for each comparison seems surprising. Effect sizes, t-scores and confidence intervals need to be in the text, and the Supplement should include a complete list of the gray and white volumes at the 6, 12 and 24 month ages for each individual 22 ASD subjects and 73 control subjects.”*

Response: Thank you for your insightful suggestions. We have taken the reviewer's advice and calculated the growth rate of GM and WM volumes from 6 to 24 months. Our analysis showed that cerebellar GM volume increased by 36.08% in autistic subjects and 34.88% in NCs, while cerebellar WM volume increased by 55.97% in autistic subjects and 55.37% in NCs over the same period. Moreover, we found that cerebellar GM volume increased by 23.34% and 10.29%, respectively, during 6→12 months and 12→24 months, while cerebellar WM volume increased by 33.53% and 16.81%, respectively, during the same periods. Our results show that not only are cerebellar GM and WM volumes larger in males with autism compared to normal controls from 6 to 24 months, but they also exhibit slightly faster growth rates. In the revised manuscript, we have included the new content in Section V-E:

“Figure 6c-d display longitudinal trajectories of GM and WM from 6 months to 24 months for both autistic and typically developing NC subjects. A comprehensive list of gray and white matter volumes, along with related clinical measures for 22 autistic and 73 NC subjects, is available in Table SV and Table SVI of the Supplementary Materials. The GM volume of (autistic subjects, NCs) increases by (23.34%, 21.17%) from 6→12 months, (10.29%, 11.34%) from 12→24 months, and (36.08%, 34.88%) from 6→24 months. The WM volume of (autistic subjects, NCs) increases by (33.53%, 32.38%) from 6→12 months, (16.81%, 17.38%) from 12→24 months, and (55.97%, 55.37%) from 6→24 months of age. Although there is no significant difference between the autistic and NC groups (as shown in Table III of the Supplementary Materials), the GM growth rates from 6 to 12 months and from 12 to 24 months show small effect sizes between the two groups (6→12: Cohen's $d=0.3473$; 12→24: Cohen's $d=0.4116$). Thus, the GM and WM volumes are not only larger in males with autism compared to NC from 6 to 24 months but also have a slightly faster growth rate from 6 to 12 months.”

Furthermore, we have added a new Table SIII in the *Supplementary Materials*, which provides p -values and effect sizes for the volume differences between autistic subjects and normal control subjects. To verify the distribution of different tissue volumes, we first used the Jarque-Bera test, which revealed that autistic gray matter volumes at 12 months and 24 months were not normally distributed. Next, we used both Wilcoxon rank sum test and t -test to calculate the significant differences in tissue volumes between the groups, and found that autistic gray matter volumes showed a significant difference compared to NCs at 12 months, with a very large effect size (Cohen's $d=1.3151$). In the revised manuscript, we have added the corresponding content in Section V-E: *“To test the statistical difference, we used Wilcoxon rank sum test and t -test to calculate the significant difference of tissue volumes between the autistic and NC groups. Both tests demonstrate that the autistic GM volumes have a significant difference from those of NCs at 12 months, with a very large effect size (Cohen's $d=1.3151$)”.*

Lastly, we have redrawn Fig. 1e and Fig. 6 to include 95% confidence intervals (dashed lines) for the fitting curves. Additionally, we have included a new Table SV and Table SVI in the *Supplementary Materials*. These tables provide cerebellar GM and WM volumes (in mm^3) for both 22 autistic subjects and 73 NC subjects at 6, 12, and 24 months of age. We have also included information about each subject's sex, race,

and related clinic measures such as Mullen and ADOS. You can refer to Comment #3 of Reviewer 1 for a detailed look at the new Table SV and Table SVI of the *Supplementary Materials*.

The newly added Table SIII and redrawn Fig. 1e and Fig. 6 are pasted below for reference.

TABLE SIII
STATISTICAL DIFFERENCE FOR SUBJECTS USED IN CROSS-SECTIONAL ANALYSIS AND LONGITUDINAL ANALYSIS.

Cross-sectional Analysis (BCP, 174 normal subjects: 78 male subjects vs. 96 female subjects)												
	Cerebellar GM						Cerebellar WM					
	≤3 months	6 months	9 months	12 months	18 months	24 months	≤3 months	6 months	9 months	12 months	18 months	24 months
Normal distribution ¹ ?	✓	✗	✓	✓	✓	✓	✗	✗	✓	✓	✓	✓
p -values ²	0.1048	1.75E-04*	0.0168*	0.0526	8.46E-04*	0.0067*	0.0112*	0.0012*	0.1052	0.0433*	3.05E-04*	0.4081
p -values ³	0.0825	0.0015*	0.0131*	0.0127*	1.65E-04*	0.0121*	0.0212*	0.0011*	0.1079	0.0307*	7.58E-05*	0.2909
Cohen's d	0.7342	2.0413	1.0541	1.3476	1.3654	0.6854	1.0883	2.1281	0.6549	1.1847	1.4480	0.2462

Longitudinal Analysis (NDAR, 95 male subjects: 22 autistic subjects vs. 73 NCs)												
	Cerebellar GM			Growth Rate (Cerebellar GM)			Cerebellar WM			Growth Rate (Cerebellar WM)		
	6 months	12 months	24 months	6→12	12→24	6→24	6 months	12 months	24 months	6→12	12→24	6→24
Normal distribution ¹ ?	✓	✗	✗	✓	✓	✗	✓	✓	✓	✓	✓	✓
p -values ²	0.1760	0.0157*	0.0415*	0.1517	0.1055	0.4428	0.1341	0.0130*	0.0264*	0.3543	0.5574	0.6914
p -values ³	0.2121	0.0445*	0.1080	0.1203	0.1310	0.4787	0.1483	0.0513	0.0805	0.4150	0.4437	0.7506
Cohen's d	0.1874	1.3151	0.2813	0.3473	0.4116	0.1561	0.2443	0.3774	0.3199	0.1985	0.1958	0.0776

	Normalized Cerebellar GM			Normalized Cerebellar WM			Cerebellum			Normalized Cerebellum		
	6 months	12 months	24 months	6 months	12 months	24 months	6 months	12 months	24 months	6 months	12 months	24 months
Normal distribution ¹ ?	✓	✓	✓	✓	✓	✗	✓	✓	✓	✓	✓	✓
p -values ²	0.2759	0.5634	0.456	0.4298	0.7077	0.9262	0.1647	0.0162*	0.0265*	0.7408	0.8982	0.7011
p -values ³	0.2700	0.5079	0.3212	0.5460	0.8244	0.9662	0.1682	0.0374*	0.0856	0.7107	0.7890	0.5779
Cohen's d	0.2735	0.1670	0.2477	0.1454	0.0550	0.0104	0.3228	0.4776	0.3907	0.0922	0.0669	0.1399

¹ Jarque-Bera test is used to test the normal distribution.

² Wilcoxon rank sum test.

³ Two-sample *t*-test.

* *p*-value < 0.05.

Fig. 1e. Cross-sectional growth trajectories based on 174 normal control infant subjects (solid lines; dashed lines: 95% confidence intervals): (left) scatterplots of gray matter and white matter volumes of infant cerebellum in the first two postnatal years; (right) scatterplots of normalized gray matter and white matter volumes in terms of the total cerebellum volume (TCV) in the first two postnatal years.

Fig. 6. **Cross-sectional analysis based on 174 NC subjects (78M/96F; Solid lines: fitted trajectories; Dashed lines: 95% confidence intervals): a**, Scatterplots of cerebellar gray matter volumes between male and female infants in the first two postnatal years. **b**, Scatterplots of cerebellar white matter volumes between male and female infants in the first two postnatal years. **Longitudinal analysis based on 95 male subjects from 6 months to 24 months (NC subjects: 73M; Autistic subjects: 22M; Solid lines: fitted trajectories; Dashed lines: 95% confidence intervals): c**, Longitudinal trajectories of cerebellar gray matter volumes for male autistic infants and NC from 6 months to 24 months. **d**, Longitudinal trajectories of cerebellar white matter volumes for male autistic infants and NC from 6 months to 24 months. The *p*-values and effect sizes are available at Table SIII of the *Supplementary Materials*.

Comment 4. “In the previous review, the suggestion was to add motivation and literature background about the cerebellum in ASD. The revision response seems to be that the sample of 22 ASD subjects is added in order to advance knowledge that could impact early diagnosis. However, virtually no one in the field sees MRI of ASD as ever becoming a diagnostic tool. Previous claims that ASD MRI measures can be accurate diagnostic classifiers are based on deeply flawed classifier discovery methodology. The motivation should be that more accurate MRI measures at early ages will advance insight into the neural and biological bases of developmental disorders such as ASD, not clinical diagnosis. Regarding the background literature, the authors did add two articles which is fine, but the ASD cerebellum literature is really quite large and is insufficiently represented. What is relevant to state is that, whereas a small number of past studies separately quantified gray and white volumes at very ages in ASD, most previous ASD studies quantified only whole cerebellum volumes at older ages (children, adolescents, adults) in largely high ability ASD subjects. Older and higher ability ASD subjects may be less likely to display cerebellar differences from controls, while lower ability infants and toddlers may have stronger cerebellar developmental differences from typicals. For that reason, the manuscript focus on early age methodology is very important, and the very young $N=22$ ASD subjects are a good choice as a proof of principle (but the subjects need to be better clinically characterized in the manuscript).”

Response: Thank you for your insightful suggestions and valuable comments. In response to the reviewer's comments, we have revised the motivation of our study as “Therefore, investigating the early development of the cerebellum, particularly during the first two postnatal years, is crucial for gaining a better understanding of cerebellar structure and function. Accurate neuroimaging measures can provide further insights into the neural and biological bases of developmental disorders” (Section I), and “accurate neuroimaging measures at early ages will further advance insight into the neural and biological bases of the developmental disorders” (Section V-E). To provide more detailed background, we reviewed 50+ related articles and included 10+ representative research articles on volumetric analysis of autistic brains. To the best of our knowledge, there has been no longitudinal voxel-based volumetric analysis of autistic cerebellar tissues (gray and white matter) during

the first two postnatal years. In the revised manuscript, we have updated the corresponding content in Section V-E:

“Early detection of autism through biomarkers is highly desirable, as it enables timely intervention and positive long-term effects on symptoms. However, due to the unclear neuroanatomy of autism, diagnosis typically occurs late, around the age of 2 [49]. To identify potential biomarkers, numerous neuroimaging studies have been proposed to compare brain structures of autistic subjects with normal controls (NCs). Voxel-based volumetric analysis has revealed that autistic subjects have a consistently larger total brain volume than age-matched NC subjects across various age ranges, such as 1.9-5.2 years [50], 6-22+ years [51], and 6-35 years [52]. Other features found in autistic subjects include reduced cortical white matter volume (6-35 years [52]), increased ventricular volume (6-65 years [53]), decreased vermis volume (7.5-18.5 years [54]), increased total cerebellar volume (10-30 years [55]), increased cerebellar white matter volume (1.9-5.2 years [50]), and reduced cerebellar gray matter volume (8-13 years [56]). From the longitudinal analysis [52] on subjects aged 6-35 years, volumetric trajectories of the total cerebellar volumes may follow inverted-U curves, which show increased volumes in young children with autism and subsequently decreased to meet the curve of NC groups at 12-13 years of age.

However, most existing studies have either focused on subjects from early childhood to young adulthood or infant subjects without differentiating between gray matter and white matter in cerebellar MRIs. This may be due to the lack of reliable cerebellum segmentation tools. Recent evidence has highlighted the consistent impact of autism on the cerebellum [50], [57-60]. Nonetheless, our understanding of the cerebellar growth trajectory in the early postnatal stages of autistic infants remains extremely limited. To address this gap, with the reliable and accurate tissue segmentations generated by the proposed method, we investigate potential differences in cerebellar growth trajectories between male autistic and NC subjects.”

Comment 5. *“These comments can be easily and quickly addressed, and they do not diminish the value of the new precision method, which can stand on its own as meriting publication in Nature Communication. Once addressed, I recommend accepting the manuscript for publication.”*

Response: We greatly appreciate your approval of our research work and would like to express our gratitude for your constructive suggestions. We have thoroughly addressed the concerns according to your comments and made the necessary revisions to improve the rigor and integrity of our study. Thank you again for your valuable comments.

1. Zöllei, L., et al., *Infant FreeSurfer: An automated segmentation and surface extraction pipeline for T1-weighted neuroimaging data of infants 0-2 years*. Neuroimage, 2020. **218**: p. 116946.
2. Chen, J., et al. *Automatic Accurate Infant Cerebellar Tissue Segmentation with Densely Connected Convolutional Network*. in *Machine Learning in Medical Imaging*. 2018. Cham: Springer International Publishing.
3. Wang, L., et al. *Volume-Based Analysis of 6-Month-Old Infant Brain MRI for Autism Biomarker Identification and Early Diagnosis*. in *Medical Image Computing and Computer Assisted Intervention – MICCAI 2018*. 2018. Cham: Springer International Publishing.
4. Romero, J.E., et al., *CERES: A new cerebellum lobule segmentation method*. Neuroimage, 2017. **147**: p. 916-924.
5. Brebisson, A. and G. Montana, *Deep Neural Networks for Anatomical Brain Segmentation*. 2015.
6. Henschel, L., et al., *FastSurfer - A fast and accurate deep learning based neuroimaging pipeline*. Neuroimage, 2020. **219**: p. 117012.
7. Li, W., et al. *On the Compactness, Efficiency, and Representation of 3D Convolutional Networks: Brain Parcellation as a Pretext Task*. in *Information Processing in Medical Imaging*. 2017.
8. Wang, H. and P.A. Yushkevich, *Multi-atlas segmentation with joint label fusion and corrective learning-an open source implementation*. Front Neuroinform, 2013. **7**: p. 27.
9. Fischl, B., *FreeSurfer*. Neuroimage, 2012. **62**(2): p. 774-81.
10. Huo, Y., et al., *3D whole brain segmentation using spatially localized atlas network tiles*. NeuroImage, 2019. **194**: p. 105-119.

11. Coupé, P., et al., *AssemblyNet: A large ensemble of CNNs for 3D Whole Brain MRI Segmentation*. 2019.

NCOMMS-21-49431A-Z

Manuscript Title: Self-Supervised Learning with Application for Infant Cerebellum Segmentation and Analysis

As I wrote in my previous review, this self-supervised learning method for segmenting gray and white matter in the 0 to 24 month old cerebellum is the most sophisticated of its type to date, and the effort is large. It is head and shoulders above previous efforts to perform precision measurements of gray and white matter in the cerebellum during early development.

I read the revised manuscript and the authors' responses to my previous questions and suggestions, and I am satisfied with them. The authors have added substantive information via tables and text regarding previous reviewer comments. I recommend accepting the manuscript for publication.

Minor points to fix:

Perhaps revise several sentences in the Abstract as follows:

~~“This represents one of the first ~~tempts~~ attempts to investigate human cerebellum growth from birth to 2 years. Our ~~finding~~ findings indicate that the first six months ~~may be the most~~ is a rapid and dynamic period of growth.”~~

~~“We also find that ~~both~~ GM and WM volumes are larger in males than females, and that GM and WM volumes ~~are~~ were larger in a small sample of autistic males than ~~controlled~~ males.”~~

~~“Although these findings are limited by the number of subjects, ~~and will advance insight into the neural and biological bases of developmental disorders.~~ The new cerebellar segmentation method we describe will be valuable for future studies of cerebellar structure and function in typical and disordered development.”~~